# Realization of the Neural Fuzzy Controller for the Sensorless PMSM Drive Control System

**Hung-Khong Hoai [1,2] , Seng-Chi Chen [1,*] and Chin-Feng Chang [3]**

[1] Department of Electrical Engineering, Southern Taiwan University of Science and Technology, Tainan City 71005, Taiwan; da62b201@stust.edu.tw

[2] Faculty of Electrical and Electronics Engineering, Ho Chi Minh City University of Transport, Ho Chi Minh City 70000, Vietnam

[3] Fukuta Electric & Machinery Co., Ltd., Taichung City 429, Taiwan; gordon.chang@fukuta-motor.com.tw

[*] Correspondence: amtfcsg123@stust.edu.tw; Tel.: +886-6-253-3131 (ext. 3324)

**Abstract:** A neural fuzzy controller (NFC)-based speed controller for the sensorless permanent magnet synchronous motor (PMSM) drive control system is realized in this paper. The NFC is a fuzzy logic controller (FLC), which adjusts the RBFNN-based (radial basis function neural network) parameter by adapting the dynamic system characteristics. For sensorless PMSM drive, the integration of sliding mode observer (SMO) and phase-locked loop (PLL) is executed to estimate the rotor position and speed. To eliminate the initial rotor position estimation and overcome the conventional PLL-based position estimation error in the direction reversion transition, the *I-f* control strategy is applied to start up the motor and change the rotational direction effectively. The system performance was verified in various experimental conditions. The simulation and experimental results indicate that the proposed control algorithm is implemented efficiently. The motor starts up with diverse external loads, operates in a wide speed range for both positive and negative directions, and reverses the rotational direction stably. Furthermore, the system presents robustness against disturbance and tracks the command speed properly.

**Keywords:** permanent magnet synchronous motor; sliding mode observer; sensorless control; neural fuzzy controller; *I-f* control strategy

## 1. Introduction

Recently, permanent magnet synchronous motors (PMSMs) have been applied in industrial applications and electric vehicles because of their outstanding specifications such as high efficiency, high power density, and large torque-to-weight ratio. In these applications, the field-oriented control (FOC) algorithm is executed effectively, requiring precise rotor position and speed, which are usually measured by a mechanical position sensor installed on the motor's shaft such as digital encoder, resolver, and Hall sensor. However, the position sensor installation increases the size and cost for the PMSM drive control system. In some cases, sensor failure makes the PMSM system unstable and reduces control reliability. To solve these shortcomings, sensorless control techniques have been studied. Since the back electromotive force (EMF) provides the information of rotor position and speed, back EMF-based estimators are widely proposed, such as the Luenberger observer [1,2], the extended Kalman filter (EKF) observer [3–5], the model reference adaptive system (MRAS) observer [6,7], or the sliding mode observer (SMO) [8–15]. Among those observers, SMO is the most applicable because it has simple structure, robustness against disturbance, low sensitivity to parameter perturbations, and easy implementation.

Generally, the arctangent function is realized to get the rotor position information based on the estimated back EMF obtained by the SMO solution. However, the back EMF is impacted by the chattering problem and noise. To increase the estimator's accuracy and robustness, several phase-locked loop (PLL) structures have been widely researched [11–13,15–18]. The conventional PLL is suitable for one-direction rotation motor only when the parameters are already configured. The reason is that it is utilized with an estimation error, influenced by the sign of speed in the estimated back EMF, which makes a large error of 180° in rotor position estimation when the direction reversion transition happens. Thus, the conventional PLL is not applied in some applications, requiring reversible motor rotational direction operation. To overcome the disadvantage of the conventional PLL, the tangent function-based estimation error is introduced in [16]. However, it is very sensitive to noise, and the obvious estimation error may be obtained when the back EMF crosses zero. Other research using the back EMF signal reconstruction-based estimation error is presented in [12,18]. Although two abovementioned solutions can solve the limitation of the conventional PLL, the position estimation error is not guaranteed to converge to zero because they have three stable points in the phase portrait. Additionally, the estimation error in three aforementioned structures of PLL is constructed from the estimated back EMF, which is too small at the standstill stage or the low-speed region and is affected by noise significantly. Furthermore, to ensure the estimator works precisely, the motor must be ramped up to the specific speed threshold, where the back EMF is large enough for estimation. Therefore, to overcome the startup and reversal problems, the combination of the *I-f* control strategy and the SMO-PLL estimator is implemented in this paper, where the conventional PLL is modified by a position offset value.

The high-performance motor drive control system requires a good tracking response, robustness against disturbance, adaption to parameter variations, and a wide adjustable speed range. Therefore, many speed controllers, such as the RBFNN (radial basis function neural network)-based self-tuning PID (Proportional-Integral-Derivative) controller [15], the sliding mode controller [19], the fuzzy neural network controller [20], and the adaptive fuzzy controller [21] have been proposed to improve system performance. Among these algorithms, in this paper, the neural fuzzy controller (NFC) researched in [22] with only the simulation results for the sensor-based PMSM drive control system is presented and realized in the real-time hardware platform. The main core of NFC is the fuzzy logic controller (FLC). FLC is one of the most effective approaches for controlling the nonlinear system, converting the linguistic control rules based on experts' knowledge into an inference mechanism to regulate the control signals on the system appropriately. Although several successful industrial applications of FLC have been implemented, proper fuzzy membership functions and fuzzy rules designation without the online learning and adaptive adjustment algorithm are not easy to be executed. Thus, FLC has incorporated a neural network with an adaptive mechanism. This integrated controller inherits the merits of both the FLC and the neural network. It performs the identification and learning ability, the parameter adaptation, and uncertain information handling concurrently. In the structure of NFC, the system is firstly identified by an RBFNN which has a simple neural network structure with fast convergence property. Then, the parameters of FLC are tuned by an adjusting mechanism based on the real-time identified plant information (Jacobian transformation) and the gradient descent method to adapt to the dynamics system characteristics.

Accordingly, in our paper, an NFC-based speed controller is realized to enhance the performance of the sensorless PMSM drive control system, which is based on the combination of the SMO-PLL estimator and the *I-f* control strategy. The control algorithm was first evaluated in simulation and then deployed to a real-time platform, established on the DSP (Digital Signal Processor) F28379D (C2000™ LAUNCHXL, Texas Instruments, Inc., Dallas, TX, USA). The system performance is checked under the different experimental conditions with a dynamic load system. The analyzed results demonstrate that the system starts up with diverse external loads, operates in two directions, and switches the rotational direction stably. Comparing to the PI (Proportional-Integral) speed controller, the NFC-based speed controller presents a better tracking performance, faster response and robustness against disturbance.

Additionally, the development time for DSP application is substantially shortened when the motor control algorithm is integrated into MATLAB Simulink (MATLAB 2017a, The Mathworks, Inc., Natick, MA, USA) based on the Embedded Coder Support Package utilities.

The remainder of this paper is arranged as follows. Section 2 presents the sensorless PMSM drive control system, including the sliding mode observer, the PLL's structures, and the *I-f* control strategy. The structure of neural fuzzy controller is described in Section 3, consisting of the FLC, the RBFNN, and the adjusting mechanism. In Section 4, the effectiveness and the correctness of the control algorithm are evaluated by the simulation results. The proposed algorithm is confirmed on the real-time platform in Section 5. Finally, Section 6 gives the conclusion.

## 2. Description of the Sensorless PMSM Drive Control System

The overall architecture of the NFC-based speed controller for the sensorless PMSM drive control system is presented in Figure 1. As mentioned above, the *I-f* control mode and the sensorless control mode are investigated to control the motor in a wide speed range for both directions, which also includes starting the motor from the standstill stage and the rotational direction reversion. In the low-speed range, the PMSM is controlled in the *I-f* control mode, where the switches are activated in position 1. In the medium- to high-speed range, the motor operates in the sensorless control mode, where the switches are activated in position 2. For the sensorless control, the control algorithm includes two closed control loops—the outer speed control loop and the inner current control loop. The FOC algorithm and two PI controllers are implemented in the current control loop. The speed control loop is realized by an NFC-based speed controller, consisting of an FLC, an RBFNN, an adjusting mechanism, and a PI controller. The detailed control algorithm and related formulation are described as follows.

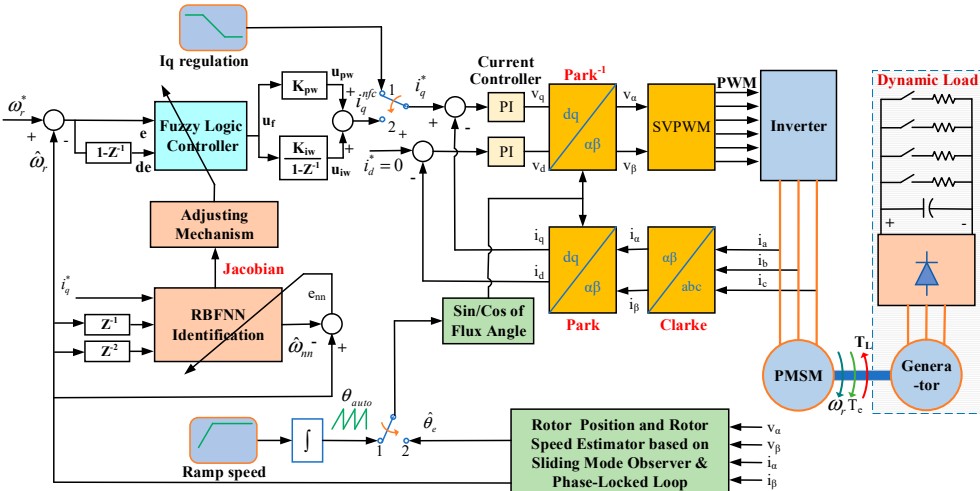

**Figure 1.** The overall architecture of the neural fuzzy controller (NFC)-based speed controller for the sensorless permanent magnet synchronous motor (PMSM) drive control system.

### 2.1. Sliding Mode Observer

The modeling of the PMSM was presented in detail previously [15]. In this paper, the main equations for PMSM and the principle of sliding mode observer are summarized as follows.

On the $\alpha$–$\beta$ stationary coordinates, the current equation of the surface-mounted PMSM (SPMSM) can be expressed by

$$\begin{cases} \frac{d}{dt}i_\alpha = \frac{1}{L_s}(-r_s i_\alpha + v_\alpha - e_\alpha) \\ \frac{d}{dt}i_\beta = \frac{1}{L_s}\left(-r_s i_\beta + v_\beta - e_\beta\right) \end{cases} \tag{1}$$

where, $i_\alpha, i_\beta, v_\alpha, v_\beta$ represent the current and the voltage of the $\alpha$ and $\beta$ axis, respectively; $e_\alpha, e_\beta$ are the back EMF, satisfying $e_\alpha = -\omega_e \lambda_f sin\theta_e, e_\beta = \omega_e \lambda_f cos\theta_e$; $\theta_e$ is the electrical rotor position; $\lambda_f$ is the

permanent magnet flux linkage; $\omega_e$ is the rotating speed of magnet flux; and $r_s$ and $L_s$ are the phase resistance and the phase inductance, respectively.

The current observer formulation for SPMSM can be formulated as follows, based on the sliding mode observer theory:

$$\begin{cases} \frac{d}{dt}\hat{i}_\alpha = \frac{1}{L_s}\left(-r_s\hat{i}_\alpha + v_\alpha - kH(\hat{i}_\alpha - i_\alpha)\right) \\ \frac{d}{dt}\hat{i}_\beta = \frac{1}{L_s}\left(-r_s\hat{i}_\beta + v_\beta - kH(\hat{i}_\beta - i_\beta)\right) \end{cases} \tag{2}$$

where $\hat{i}_\alpha, \hat{i}_\beta$ represent the estimated current of the $\alpha$ and $\beta$ axis, $H(x)$ is the sigmoid function, and $k$ is the observer gain value. Additionally, formulated in the dynamic equation, the estimated current error can be expressed by subtracting Equation (1) from Equation (2):

$$\begin{cases} \frac{d}{dt}\widetilde{i}_\alpha = \frac{1}{L_s}\left(-r_s\widetilde{i}_\alpha + e_\alpha - kH(\widetilde{i}_\alpha)\right) \\ \frac{d}{dt}\widetilde{i}_\beta = \frac{1}{L_s}\left(-r_s\widetilde{i}_\beta + e_\beta - kH(\widetilde{i}_\beta)\right) \end{cases} \tag{3}$$

where $\widetilde{i}_\alpha, \widetilde{i}_\beta$ represent the estimated current error of the $\alpha$ and $\beta$ axis, defined as: $\widetilde{i}_\alpha = \hat{i}_\alpha - i_\alpha$ and $\widetilde{i}_\beta = \hat{i}_\beta - i_\beta$.

Furthermore, the asymptotic stability of the SMO based on the Lyapunov theory is guaranteed when the observer gain $k$ is designed to be a sufficiently large positive value, $k > max(|e_\alpha|, |e_\beta|)$. Therefore, the estimated current will converge to the actual current. The estimated back EMF $\hat{e}_\alpha, \hat{e}_\beta$ can be obtained as:

$$\begin{cases} \hat{e}_\alpha = kH(\hat{i}_\alpha - i_\alpha) = kH(\widetilde{i}_\alpha) \\ \hat{e}_\beta = kH(\hat{i}_\beta - i_\beta) = kH(\widetilde{i}_\beta) \end{cases} \tag{4}$$

### 2.2. Speed and Position Estimation Method

Based on the estimated back EMF, the rotor position is directly determined by the arctangent function in the traditional method

$$\hat{\theta}_e = arctan\left(\frac{\hat{e}_\alpha}{\hat{e}_\beta}\right) \tag{5}$$

and the electrical rotor speed can be obtained by $\hat{\omega}_e = \frac{d\hat{\theta}_e}{dt}$. However, due to using the arctangent function, the estimated position and speed are vulnerable to noise and harmonics. Particularly, the apparent estimation errors can be obtained when $\hat{e}_\beta$ crosses zero. Therefore, the phase-locked loop algorithm is executed to mitigate the adverse effect. The structure of the PLL is shown in Figure 2.

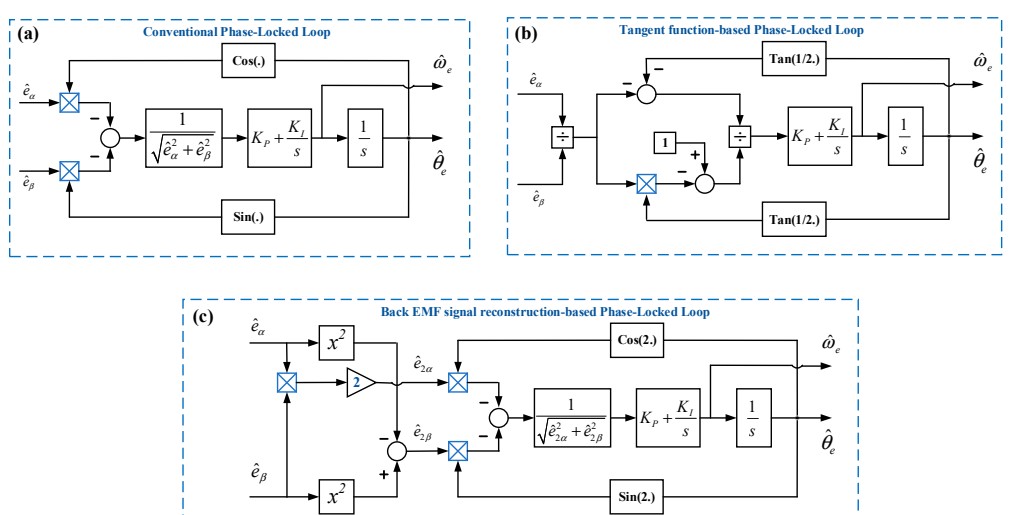

**Figure 2.** The structures of the PLL (phase-locked loop).

In the conventional PLL [15], the estimation error of rotor position is expressed by

$$
\begin{aligned}
\varepsilon &= \frac{1}{\sqrt{\hat{e}_\alpha^2 + \hat{e}_\beta^2}}\Big[-\hat{e}_\alpha cos(\hat{\theta}_e) \ - \ \hat{e}_\beta sin(\hat{\theta}_e)\Big] \\
&= -\hat{e}_{\alpha n}cos(\hat{\theta}_e) \ - \ \hat{e}_{\beta n}sin(\hat{\theta}_e) \\
&= sin(\theta_e)cos(\hat{\theta}_e) \ - \ cos(\theta_e)sin(\hat{\theta}_e) \\
&= sin(\theta_e - \hat{\theta}_e) \approx \ \theta_e - \hat{\theta}_e
\end{aligned}
\tag{6}
$$

In the tangent function-based PLL [16], the estimation error of rotor position is expressed by

$$
\varepsilon = \frac{-\frac{\hat{e}_\alpha}{\hat{e}_\beta} - tan\left(\frac{\theta_e}{2}\right)}{1 - \frac{\hat{e}_\alpha}{\hat{e}_\beta}tan\left(\frac{\hat{\theta}_e}{2}\right)} = \frac{tan(\theta_e) - tan\left(\frac{\theta_e}{2}\right)}{1 + tan(\theta_e)tan\left(\frac{\hat{\theta}_e}{2}\right)} = tan\left(\theta_e - \frac{\hat{\theta}_e}{2}\right) .
\tag{7}
$$

In the back EMF signal reconstruction-based PLL [12,18], the estimation error of rotor position is expressed by

$$
\begin{aligned}
\varepsilon &= \frac{1}{\sqrt{\hat{e}_{2\alpha}^2 + \hat{e}_{2\beta}^2}}\Big[-\hat{e}_{2\alpha}cos(2\hat{\theta}_e) \ - \ \hat{e}_{2\beta}sin(2\hat{\theta}_e)\Big] \\
&= sin\big[2\big(\theta_e - \hat{\theta}_e\big)\big]
\end{aligned}
\tag{8}
$$

where $\hat{e}_{2\alpha}$, $\hat{e}_{2\beta}$ are the modified back EMF signal, defined as: $\hat{e}_{2\alpha} = 2\hat{e}_\alpha\hat{e}_\beta = -\big(\omega_e\lambda_f\big)^2 sin(2\theta_e)$ and $\hat{e}_{2\beta} = \hat{e}_\beta^2 - \hat{e}_\alpha^2 = \big(\omega_e\lambda_f\big)^2 cos(2\theta_e)$.

The phase portrait of the PLLs is presented in Figure 3. In the tangent function-based PLL and back EMF signal reconstruction-based PLL, there are three stable points for both positive and negative rotor speed, which are (0,0), ($\pi$,0) and ($-\pi$,0). When the PI regulator's parameters are tuned properly, the rotor position estimation error can approach to the origin point. However, in practical application, when the motor changes its direction, the back EMF is very small and is affected by noise significantly in the low-speed region, which causes a large estimation error. Moreover, depending on the initial rotor position and speed estimation error, the trajectories can converge to the stable point (0,0) or ($\pm\pi$,0) when the motor starts up from the standstill stage. Especially, in the case of the *I-f* startup strategy without initial rotor position estimation, the estimation error uncertainly converges to the origin point. It implies that the position estimation error can be obtained of 0 or $\pi$ (rad). Otherwise, the cycle of rotor position is $2\pi$; thus, the conventional PLL is considered to always have only one stable point in each direction. In the conventional PLL, when the motor reverses its direction from the positive rotor speed to the negative rotor speed, the stable point (0,0) changes into the saddle point while the saddle points ($\pm\pi$,0) become stable points. This shows that, when the speed reversion transition occurs, the trajectories in the phase portrait of the conventional PLL system will deviate from (0,0) to reach ($\pm\pi$,0), and cause a rotor position estimation error of $\pi$ (rad). To solve this problem, the PI regulator's parameters should be reset. However, it is not easy to adjust to the real-time control system. To overcome the abovementioned problems, during the direction reversion transition, the *I-f* control strategy in the combination with the modified conventional PLL is studied to make sure that the motor can switch the rotational direction stably, and the conventional PLL can work effectively for both directions. That also rejects the effect of the back EMF estimation error in the low-speed region.

Therefore, a position offset is considered to modify the estimated rotor position. The estimation error of rotor position for the conventional PLL is rewritten by

$$\varepsilon = \frac{1}{\sqrt{\hat{e}_\alpha^2 + \hat{e}_\beta^2}}\left[-\hat{e}_\alpha cos(\hat{\theta}_e + \theta_{offset}) - \hat{e}_\beta sin(\hat{\theta}_e + \theta_{offset})\right]$$

$$with \ \theta_{offset} = \begin{cases} 0 & if \ \hat{\omega}_e \geq 0 \\ \pi & if \ \hat{\omega}_e < 0 \end{cases}$$

(9)

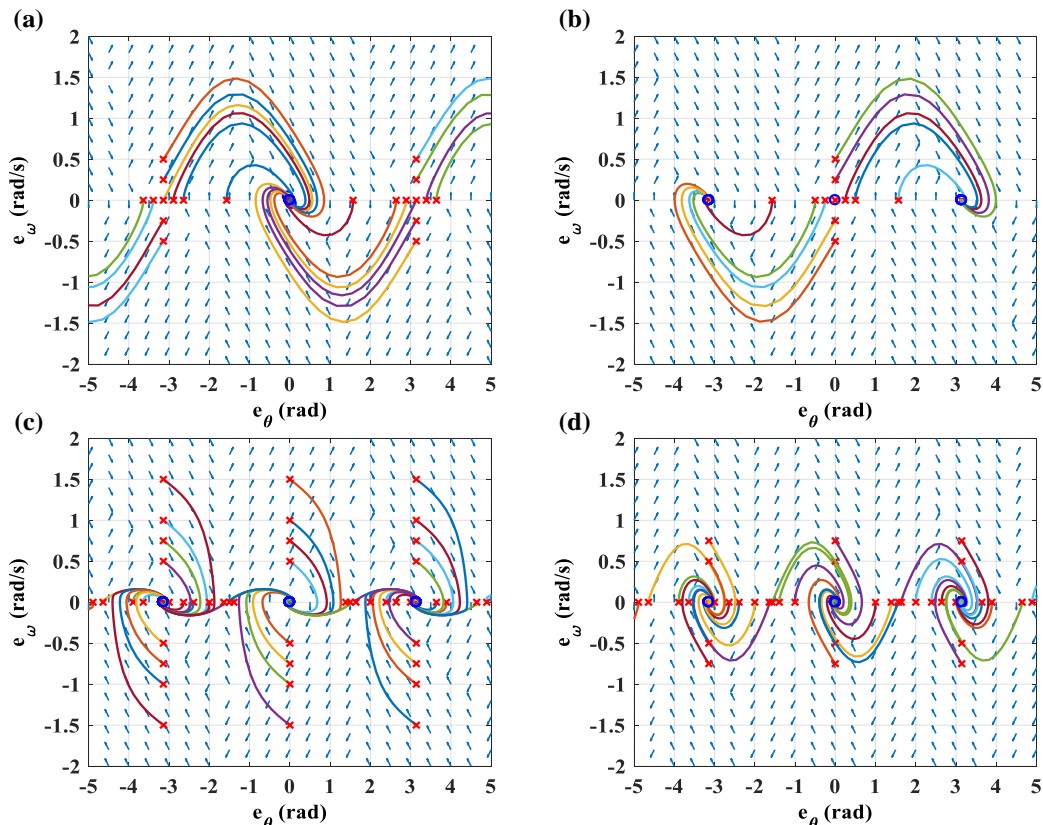

**Figure 3.** The phase portrait of: (**a**) the conventional PLL for positive rotor speed; (**b**) the conventional PLL for negative rotor speed; (**c**) the tangent-based PLL for both positive and negative rotor speed; and (**d**) the back EMF (electromotive force) signal reconstruction-based PLL for both positive and negative rotor speed.

If the estimation error is attracted to zero by a PI regulator, the rotor position and speed are estimated by

$$\begin{cases} \hat{\omega}_e = K_P\varepsilon + K_i \int \varepsilon dt \\ \hat{\theta}_e = \int \hat{\omega}_e dt \end{cases}.$$

(10)

## 2.3. The I-f Control Strategy for the Direction Reversion Transition

The *I-f* startup strategy was described and implemented previously [15,23], which smooths the torque and speed transition during the inter-mode transition from the startup stage to the sensorless control stage. Furthermore, for the direction reversion transition, the *I-f* control strategy is extended to make sure that the motor can switch the rotational direction stably. This procedure is presented in Figure 4 and portrayed as the following steps:

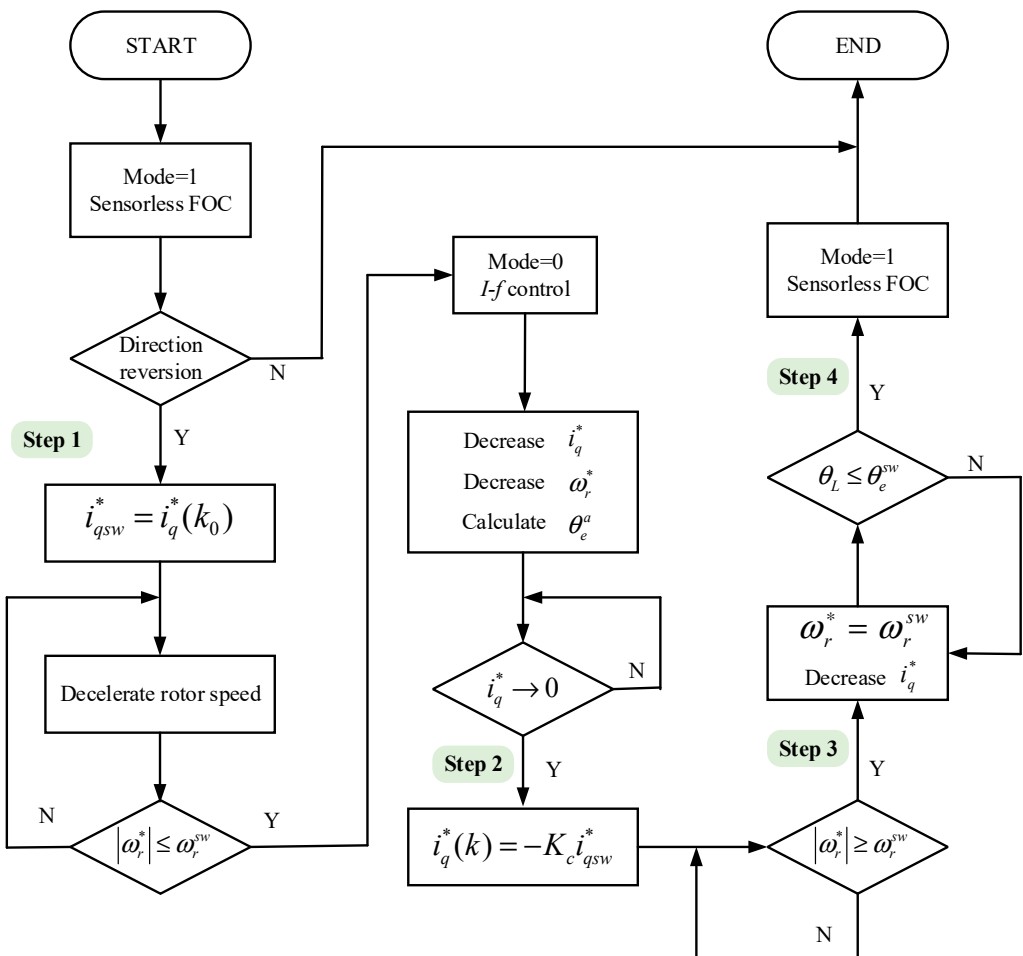

**Figure 4.** The *I-f* control strategy flowchart for the direction reversion transition.

**Step 1:** When the speed command is set up to reverse the rotational direction, the system captures the reference current $i_q^*$ and decelerates the rotor speed until the specific speed threshold value for switching condition. When the rotor speed is lower than the abovementioned value, the back-EMF signal is not large enough and significantly affected by noise, which causes the rotor position and speed information to be obtained by the SMO-PLL estimator inaccurately. Thus, the system switches to the *I-f* control mode. The speed command is decreased by a ramp function and $i_q^*$ is regulated properly. The reference current $i_q^*$ and reference speed $\omega_r^*$ are expressed by

$$\begin{cases} i_q^*(k+1) = i_q^*(k) - K_a T_s i_{qsw}^* \\ \omega_r^*(k+1) = \omega_r^*(k) \mp K_r T_s \ with \ \left|\omega_r^*\right| < \omega_r^{sw} \qquad for \ k > k_0 \\ \theta_e^a(k+1) = \theta_e^a(k) + \omega_r^* T_s \ with \ \theta_e^a(k_0) = \hat{\theta}_e(k_0), \end{cases} \tag{11}$$

where the symbol "−" denotes the positive–negative direction transition, and the symbol "+" denotes the negative–positive direction transition; $i_{qsw}^*$ is the reference current at the time the direction reversion command is set up; $K_a$ is the proportional gain for decreasing the current $i_q^*$; $\theta_e^a$ is the auto-generated rotor position; $\omega_r^{sw}$ is the specific speed threshold value for switching to the sensorless control mode; and $K_r$ is the proportional gain of the ramp function for the command speed.

**Step 2:** When the reference current $i_q^*$ approaches zero, it is reset to the certain value to speed up the motor, while the reference speed is still changed by the ramp function.

$$i_q^*(k) = -K_c i_{qsw}^* \tag{12}$$

where $K_c$ is the proportional gain of the reference current $i_q^*$ in the new direction.

**Step 3:** When the reference speed approaches $\omega_r^{sw}$, the reference current $i_q^*$ is decreased, similar to the procedure in the *I-f* startup strategy.

$$\begin{cases} i_q^*(k+1) = i_q^*(k) \mp K_a T_s \\ \omega_r^*(k+1) = \omega_r^*(k) = \omega_r^{sw} \end{cases} \quad while \ \theta_L = \hat{\theta}_e - \theta_e^a > \theta_e^{sw} \tag{13}$$

where $\theta_e^{sw}$ is the angular threshold value of the switching condition.

**Step 4:** The motor is switched to operate in the sensorless control mode.

## 3. The Neural Fuzzy Controller Design for PMSM Drive Control System

*3.1. Fuzzy Logic Controller*

In Figure 1, there are two inputs for FLC, the rotor speed error $e$ and the rotor speed error change $de$, which are expressed by

$$\begin{cases} e(k) = \omega_r^*(k) - \hat{\omega}_r(k) \\ de(k) = e(k) - e(k-1) \end{cases} \tag{14}$$

and the output of the FLC is presented as $u_f$.

According to the fuzzy control theory, the design procedure of the FLC algorithm in this paper is depicted as follows. Firstly, the input variables are selected, where their linguistic variables are presented by $E$ and $dE$. There are seven linguistic values for each input variables, labeled as $\{A_1, A_2, A_3, A_4, A_5, A_6, A_7\}$ and $\{B_1, B_2, B_3, B_4, B_5, B_6, B_7\}$, respectively. The membership functions are constructed on the symmetrical triangular type, as shown in Figure 5. Secondly, the membership degrees for $e$ and $de$ are computed. For any input value, there are only two active linguistic values and the membership degrees are obtained by

$$\begin{cases} \mu_{A_i}(e) = \frac{e_{i+1} - e}{e_{i+1} - e_i} \ and \ \mu_{A_{i+1}}(e) = 1 - \mu_{A_i}(e) \\ \mu_{B_j}(de) = \frac{de_{j+1} - de}{de_{j+1} - de_j} \ and \ \mu_{B_{j+1}}(de) = 1 - \mu_{B_j}(de) \end{cases} \tag{15}$$

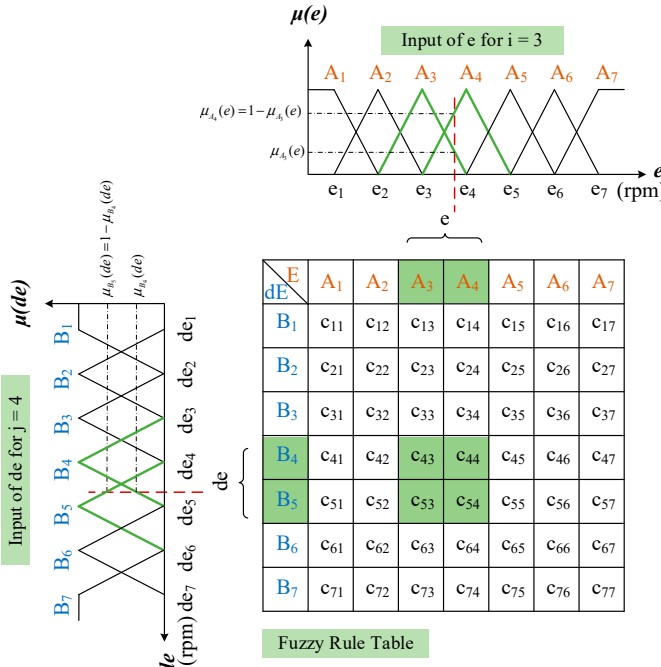

**Figure 5.** The membership function of *e, de* and fuzzy rule table.

Thirdly, the initial FLC rules are designed by referring to the dynamic response characteristics, such as

$$\text{IF } e \text{ is } A_i \text{ and } de \text{ is } B_j \text{ THEN } u_f \text{ is } c_{j,i} \tag{16}$$

where $i, j = 1, 2 \ldots 7$ and $c_{j,i}$ is the real number. There are 49 fuzzy rules in the fuzzy rule table. Finally, the product-inference rule, singleton fuzzifier, and central average defuzzification technique are implemented to infer the output $u_f(e, de)$ of the fuzzy system. For any input value $(e, de)$, the value of $(i,j)$ is determined, and only four fuzzy rules result in a non-zero output. Therefore, four linguistic values $A_i$, $A_{i+1}$, $B_j$, and $B_{j+1}$ and the corresponding consequent values $c_{j,i}$, $c_{j+1,i}$, $c_{j,i+1}$, and $c_{j+1,i+1}$ are active, and the output of the fuzzy system can be obtained by

$$u_f(e, de) = \frac{\sum_{n=i}^{i+1} \sum_{m=j}^{j+1} c_{m,n} \lfloor \mu_{A_n}(e) \times \mu_{B_m}(de) \rfloor}{\sum_{n=i}^{i+1} \sum_{m=j}^{j+1} \mu_{A_n}(e) \times \mu_{B_m}(de)} = \sum_{n=i}^{i+1} \sum_{m=j}^{j+1} c_{m,n} \times d_{m,n} \tag{17}$$

where $d_{m,n} = \mu_{A_n}(e) \times \mu_{B_m}(de)$, and the value of the $\sum_{n=i}^{i+1} \sum_{m=j}^{j+1} d_{m,n} = 1$ in Equation (17) can be easily derived by using Equation (15). Moreover, the $c_{m,n}$ are adjustable parameters of the FLC.

Additionally, in Figure 1, for the sensorless control mode, the reference current $i_q^*$ is expressed by the output of the fuzzy controller $u_f$ as the following equation:

$$i_q^*(k) = u_{iw}(k-1) + \left(K_{pw} + K_{iw}\right) \times u_f(k) \tag{18}$$

where $K_{pw}$ and $K_{iw}$ are the proportional and integral gain of PI controller in the speed control loop, respectively.

### 3.2. Radial Basis Function Neural Network

The parameters of FLC in the NFC-based speed controller should be adjusted effectively. That needs the Jacobian transformation obtained from the dynamic system characteristics. Therefore, the PMSM drive control system must be identified. In Figure 6, an RBFNN is designed to acquire that information. It is a feedforward neural network architecture comprised of an input layer, a hidden layer, and a single output layer.

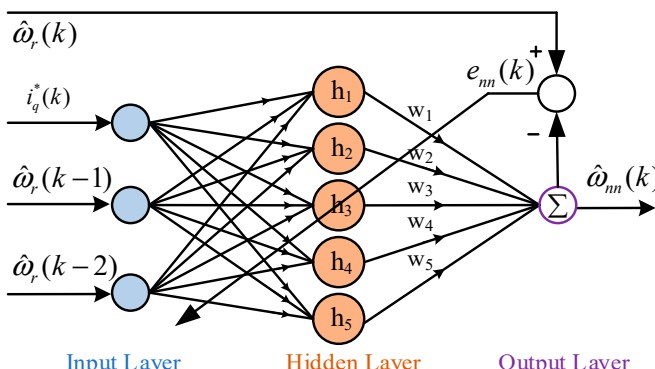

**Figure 6.** The structure of RBFNN (radial basis function neural network) in the NFC-based speed controller.

There are three inputs in the input layer, namely $i_q^*(k)$, $\hat{\omega}_r(k-1)$, $\hat{\omega}_r(k-2)$, and the input vector form is expressed by

$$x(k) = \left[i_q^*(k) \ \hat{\omega}_r(k-1) \ \hat{\omega}_r(k-2)\right]^T \tag{19}$$

In the hidden layer, there are five neurons and the Gaussian function is used as the activation function, which is presented by the following equation

$$h_l(k) = exp\left(-\frac{\|x(k) - C_l(k)\|^2}{2b_l^2(k)}\right) \tag{20}$$

where $C_l(k) = [c_{1l}(k) \ c_{2l}(k) \ c_{3l}(k)]^T$ and $c_{rl}$, $b_l$, are the center and width of the Gaussian function.

In the output layer, the output is formulated by a linearly weighted sum of hidden nodes:

$$\hat{\omega}_{nn}(k) = \sum_{l=1}^{5} w_l(k)h_l(k) \tag{21}$$

Additionally, to adjust the parameters of RBFNN by the online learning law, the instantaneous cost function is presented by

$$J(k) = \frac{1}{2}(\hat{\omega}_r(k) - \hat{\omega}_{nn}(k))^2 = \frac{1}{2}e_{nn}^2(k) \tag{22}$$

According to the stochastic gradient descent (SGD) method, the weights, node widths, and node centers can be calculated and updated by the learning algorithm as below equations [24]:

$$\begin{cases} \Delta\omega_l(k) = -\eta\frac{\partial J(k)}{\partial\omega_l(k)} = \eta e_{nn}(k)h_l(k) \\ \omega_l(k) = \omega_l(k-1) + \Delta\omega_l(k) + \alpha(\omega_l(k-1) - \omega_l(k-2)) \end{cases} \tag{23}$$

$$\begin{cases} \Delta b_l(k) = -\eta\frac{\partial J(k)}{\partial b_l(k)} = \eta e_{nn}(k)\omega_l(k)h_l(k)\frac{\|x(k)-C_l(k)\|^2}{b_l^3(k)} \\ b_l(k) = b_l(k-1) + \Delta b_l(k) + \alpha(b_l(k-1) - b_l(k-2)) \end{cases} \tag{24}$$

$$\begin{cases} \Delta c_{rl}(k) = -\eta\frac{\partial J(k)}{\partial c_{rl}(k)} = \eta e_{nn}(k)\omega_l(k)h_l(k)\frac{x_r(k)-c_{rl}(k)}{b_l^2(k)} \\ c_{rl}(k) = c_{rl}(k-1) + \Delta c_{rl}(k) + \alpha(c_{rl}(k-1) - c_{rl}(k-2)) \end{cases} \tag{25}$$

where $\alpha$ is the momentum factor; $\eta$ is the learning rate, and $r = 1, 2, 3$; $l = 1, 2 \ldots 5$.

Additionally, the Jacobian transformation can be expressed by

$$\frac{\partial\hat{\omega}_r}{\partial i_q^*} \approx \frac{\partial\hat{\omega}_{nn}}{\partial i_q^*} = \sum_{l=1}^{5} w_l(k)h_l(k)\frac{c_{1l}(k) - i_q^*(k)}{b_l^2(k)} \tag{26}$$

### 3.3. Adjusting Mechanism of Fuzzy Logic Controller

To achieve the smallest squared error between the reference speed and the estimated rotor speed, the FLC parameters are adjusted online in the closed-loop control. Hence, the cost function can be firstly defined by

$$J_e(k) = \frac{1}{2}(\omega_r^*(k) - \hat{\omega}_r(k))^2 = \frac{1}{2}e^2(k) \tag{27}$$

Then, the NFC learning law is derived based on the gradient descent method. In the fuzzy rule table (Figure 5), parameters of $c_{m,n}$ can be updated and optimally adjusted according to

$$\Delta c_{m,n} = -\gamma\frac{\partial J_e}{\partial c_{m,n}} \tag{28}$$

where $\gamma$ is the adaptive rate. The chain rule is executed, and the partial differential equation for $J_e$ in Equation (28) can be presented by

$$\frac{\partial J_e}{\partial c_{m,n}} = -\frac{\partial J_e}{\partial \hat{\omega}_r} \frac{\partial \hat{\omega}_r}{\partial u_f} \frac{\partial u_f}{\partial c_{m,n}} \tag{29}$$

Furthermore, from Equation (15) and using the Jacobian transformation from Equation (26), the following equations are obtained:

$$\frac{\partial u_f}{\partial c_{m,n}} = d_{m,n} \tag{30}$$

and

$$\frac{\partial \hat{\omega}_r}{\partial u_f} = \frac{\partial \hat{\omega}_r}{\partial i_q^*} \frac{\partial i_q^*}{\partial u_f} \approx \left(K_{pw} + K_{iw}\right) \frac{\partial \hat{\omega}_r}{\partial i_q^*} = \left(K_{pw} + K_{iw}\right) \sum_{l=1}^{5} w_l(k)h_l(k) \frac{c_{1l}(k) - i_q^*(k)}{b_l^2(k)} \tag{31}$$

Substituting Equations (30) and (31) into Equation (29), the parameters $c_{m,n}$ of FLC defined in Equation (17) can be adjusted by the following equation:

$$\Delta c_{m,n}(k) = \gamma e(k)\left(K_{pw} + K_{iw}\right) d_{m,n} \sum_{l=1}^{5} w_l(k)h_l(k) \frac{c_{1l}(k) - i_q^*(k)}{b_l^2(k)} \tag{32}$$

with $m = j, j+1$ and $n = i, i+1$.

## 4. Simulation Results

To demonstrate the correctness of the proposed control algorithm for the PMSM drive system, the system performance was tested in both a simulation and real-time hardware within different experimental conditions. The first case is controlling the motor for two directions in the wide speed range to validate the tracking response and the estimator's correction. The estimated values were compared to the actual values, which were measured by the digital incremental encoder. The second case is the direction reversion transition to make sure that the system can switch the direction stably. The third case is starting up the motor with the different initial external load and comparing the system performance between the proposed NFC-based speed controller and the PI speed controller. That confirms the executed ability of the *I-f* startup strategy and the system's robustness against disturbance. All instances are inspected on the PMSM control system and dynamic load system, where the PMSM's parameters are listed as rated power of 750 W, rated current of 4.24 A, rated speed of 2000 rpm, phase resistance of 1.326 $\Omega$, phase inductance of 2.952 mH, back-EMF constant of 56.5 $V_{L-L}$/krpm, torque constant of 0.86 Nm/A, inertia of 3.63 Kg·cm$^2$, and pole pairs of 4.

Furthermore, the PI speed controller has fixed parameters and not flexible to the PMSM drive control system, which is a dynamic, multivariable, and nonlinear system. Especially, the PMSM drive control system is usually operated under various conditions or dynamics load. Therefore, the PI controller only works effectively at a specific operating condition, where the PI parameters are designed in correspondence to the system characteristics. For a fair comparison between the NFC-based speed controller and the PI speed controller, the system performances were analyzed with the same PI's parameters. In the third experimental condition, the PI speed controller is firstly set for the case of an external load of 50 $\Omega$, satisfying the performance criteria such as the transient response without the overshoot or undershoot, and with the short settling time and the zero steady-state error. Then, the operating condition is varied by applying two different external load values on the system to evaluate the system performance for both the PI and NFC-based speed controllers.

The NFC-based speed controller for the sensorless PMSM drive control system was completely designed in MATLAB Simulink, and the overall system is shown in Figure 7. The sampling frequency for real-time platform modeling (block A) is 20 kHz. In the motor control algorithm (block B), the sampling frequency for speed loop control is 1 kHz, while the sampling frequency for the estimator

and the current control loop is 20 kHz. The simulation data (block C) are acquired and monitored with the sampling frequency of 20 kHz. The PI controllers' parameters are set in the current control loop as $K_{Pq}$ = 0.25, $K_{Iq}$ = 0.025, $K_{Pd}$ = 0.25, $K_{Id}$ = 0.025. The NFC-based speed controller is realized in the speed control loop. For the FLC, the membership function's values are set as $E$= [ $e_1$ $e_2$ $e_3$ $e_4$ $e_5$ $e_6$ $e_7$] = [−225 −150 −75 0 75 150 225] and $dE$ = [$de_1$ $de_2$ $de_3$ $de_4$ $de_5$ $de_6$ $de_7$] = [−187.5 −125 −62.5 0 62.5 125 187.5]. The initial fuzzy rule table values are set as in Table 1. These parameters are adjusted during the operating time with the adaptive rate of 0.25. For the neural network, the initial neuron parameters are set as follows: node centers $C_l$ = [−0.5 −0.25 0.0 0.25 0.5], node widths $b$ = [0.25 0.25 0.25 0.25 0.25], connective weights $w$ = [0.00625 0.00625 0.00625 0.00625 0.00625], momentum factor $\alpha$ = 0.95, and the learning rate $\eta$ = 0.475. Moreover, the PI's parameters of the PI and NFC-based speed controller are set as $K_{pw}$ = 1.725 and $K_{iw}$ = 0.030. Additionally, the dynamic load system is modeled by a generator and the electrical load. The electrical load is comprised of a rectifier, a capacitor of 470 μF, and the resistor load bank. In the first and second experimental conditions, the motor is operated with a total resistance load of 100 Ω.

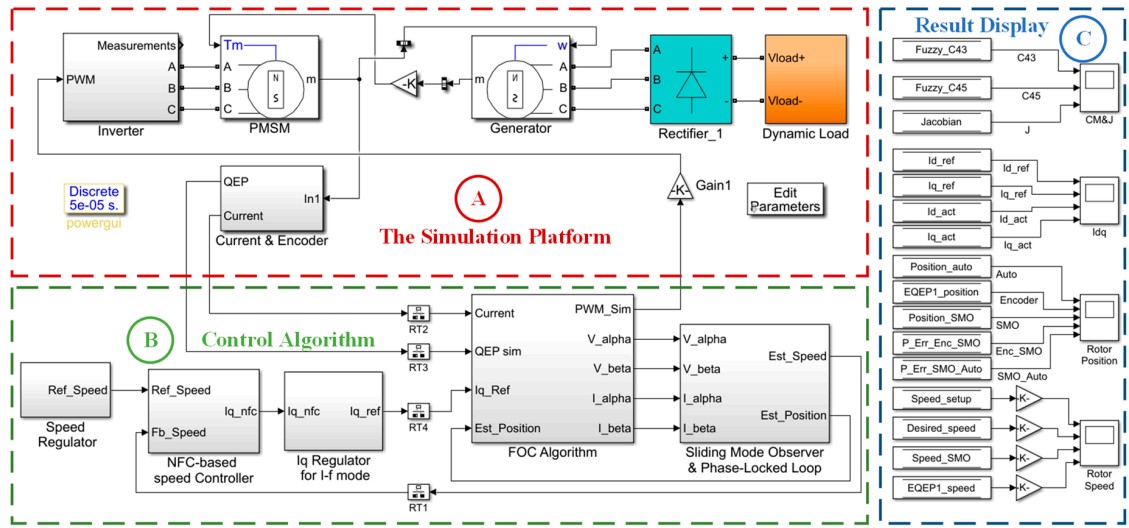

**Figure 7.** The overall simulation structure of the sensorless PMSM drive control system with a dynamic load system in MATLAB Simulink.

**Table 1.** The initial fuzzy rule table of fuzzy logic controller (FLC).

| dE \ E | $A_1$ | $A_2$ | $A_3$ | $A_4$ | $A_5$ | $A_6$ | $A_7$ |
|---|---|---|---|---|---|---|---|
| $B_1$ | −0.324 | −0.324 | −0.324 | −0.324 | −0.216 | −0.108 | 0 |
| $B_2$ | −0.324 | −0.324 | −0.324 | −0.216 | −0.108 | 0 | 0.108 |
| $B_3$ | −0.324 | −0.324 | −0.216 | −0.108 | 0 | 0.108 | 0.216 |
| $B_4$ | −0.324 | −0.216 | −0.108 | 0 | 0.108 | 0.216 | 0.324 |
| $B_5$ | −0.216 | −0.108 | 0 | 0.108 | 0.216 | 0.324 | 0.324 |
| $B_6$ | −0.108 | 0 | 0.108 | 0.216 | 0.324 | 0.324 | 0.324 |
| $B_7$ | 0 | 0.108 | 0.216 | 0.324 | 0.324 | 0.324 | 0.324 |

Figure 8 presents the motor's performance in the positive direction for the wide speed range, including: (1) the startup from the standstill; (2) stepping up to the rated speed; and (3) slowing down. The command speed is varied in a sequence of 0 → 200 → 500 → 1000 → 1500 → 2000 → 1600 → 1200 → 800 → 1000 rpm. Figure 8a indicates that the estimated rotor speed overlaps the actual value and closely tracks the command speed properly. Figure 8b implies that the current $i_q$ is regulated proportionally to the command speed while the current $i_d$ almost is equal to zero. Figure 8c–f illustrates the estimated position, actual position, and the estimation errors at the speeds of 500, 1000,

1500, and 2000 rpm in a period of 0.15 s. There are 5, 10, 15, and 20 position cycles, corresponding to the rotation frequencies of 33.33, 66.67, 100, and 133.33 Hz, respectively. These values are proper to the motor with four pole pairs. The actual and estimated positions come close to each other; thus, the estimation error is equal to zero.

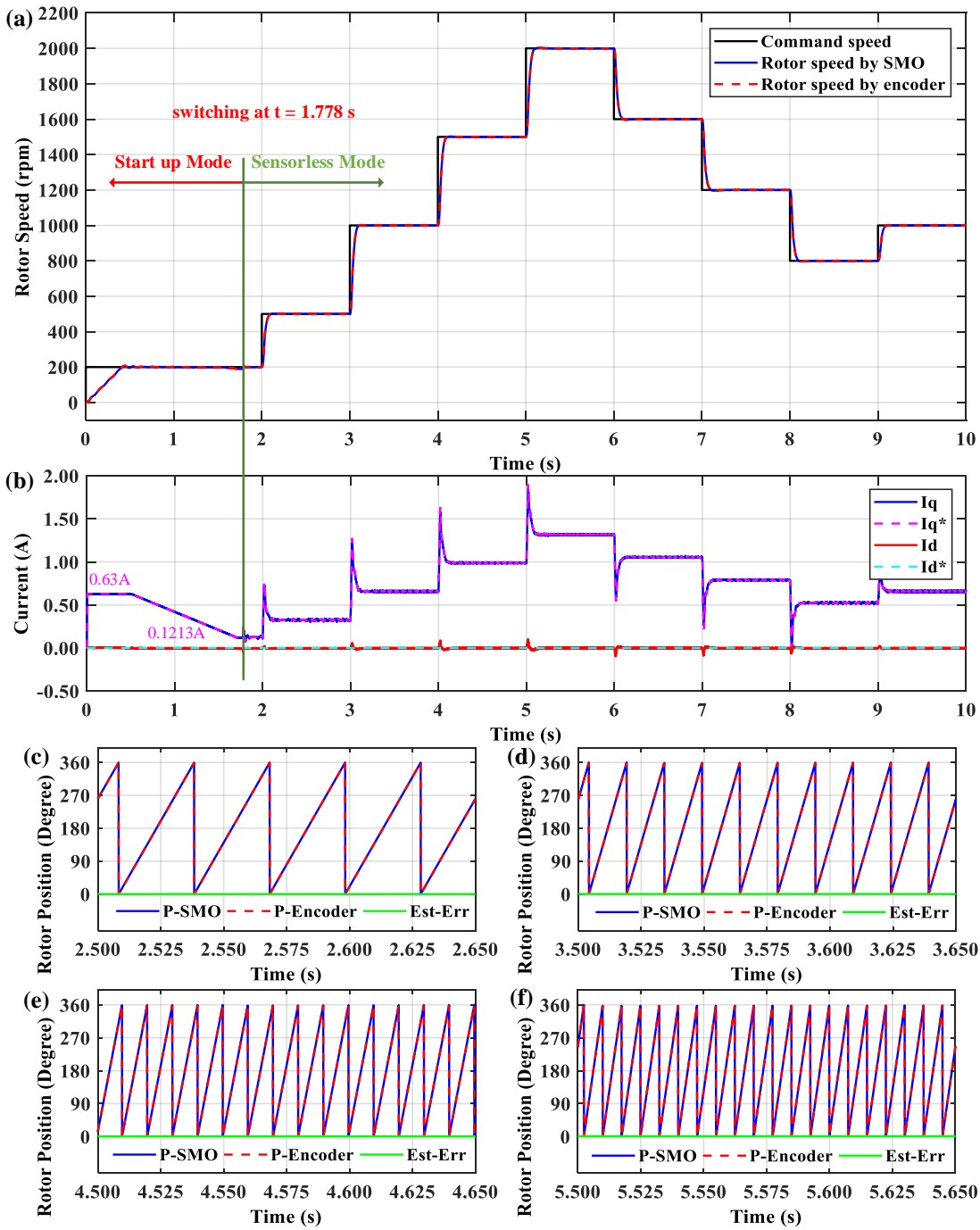

**Figure 8.** Simulation results in the wide speed range in the positive direction for: (**a**) speed response; (**b**) current response; and rotor position response at (**c**) 500; (**d**) 1000; (**e**) 1500; and (**f**) 2000 rpm.

The motor's performance in the negative direction for the wide speed range is illustrated in Figure 9. The waveform of the command speed is the same as in the positive direction, only the speed has the opposite value. Comparing to Figure 8, the speed response, the current, and the rotor position in the negative direction are just the reverse of their value. In the positive direction, the rotor position

is increased from 0° to 360° for one electrical cycle, while the rotor position is decreased from 360° to 0° in the negative direction. Therefore, Figures 8 and 9 refer that the motor control system has worked for two directions with the same quality. In both rotational directions, the motor startups from the standstill stage with the initial reference current $i_q^*$ of 0.63 A. The ratio for decreasing the reference current $i_q^*$ is set up at the value of 0.42 A/s. The motor control algorithm is switched to the sensorless control mode at t = 1.788 s, where the reference current $i_q^*$ is decreased to 0.1213 A.

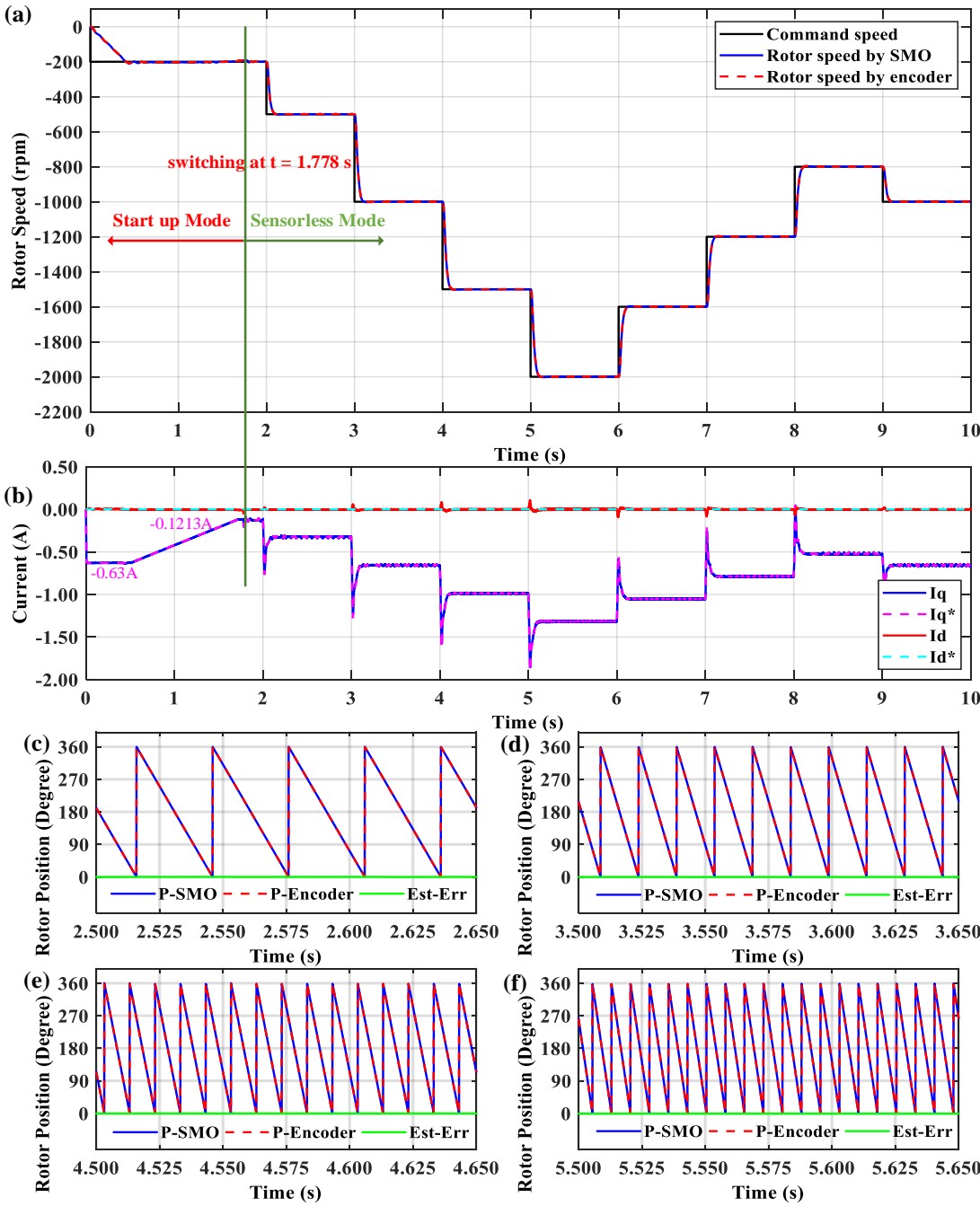

**Figure 9.** Simulation results in the wide speed range in the negative direction for: (**a**) speed response; (**b**) current response; and rotor position response at (**c**) 500; (**d**) 1000; (**e**) 1500; and (**f**) 2000 rpm.

The motor's speed performance in the reversal operation is presented in Figure 10. Firstly, the motor is started in the positive direction, and then accelerated and decelerated, following the sequence: 0 → 200 → 700 → 200 rpm. Secondly, the command speed is set up to reverse the rotational

direction from 200 to −200 rpm at t = 4 s. The motor control algorithm is transferred to the *I-f* control mode from the sensorless control mode. When the motor has already operated in the negative direction, the motor control algorithm is switched to the sensorless control mode again and the command speed is varied by the sequence: −200 → −700 → −200 rpm. Thirdly, a similar procedure is implemented when the command speed is changed to reverse the rotational direction from −200 to 200 rpm at t = 7 s. During the operation of the motor, the estimated rotor speed almost tracks the reference speed and approximates the actual rotor speed for both the sensorless control mode and the *I-f* control mode. Moreover, the current response in Figure 10b indicates that the reference current $i_q^*$ is regulated differently for the sensorless control mode and the *I-f* control mode in the rotational direction reversion. Therefore, the rising time and settling time are different in the speed transition between the rotational direction reversion and the varied speed in one direction. It takes a larger time to reverse the direction. Figure 10c illustrates the rotor position when the motor changes its direction to the negative direction; there is a larger estimation error and the maximum value is −38.67° at t = 4.077 s. Additionally, the maximum rotor position estimation error is 45.25° at t = 7.078 s in the case of changing the direction from the negative value to the positive value in Figure 10d. However, these estimation errors could be accepted because the direction reversion time only takes about 0.111 s. It is the period that the system is operated in the *I-f* control mode. Therefore, Figure 10 implies that the *I-f* control mode makes the motor change the direction stably and the modified conventional PLL works effectively. The estimation error still equals to zero after changing the rotational direction.

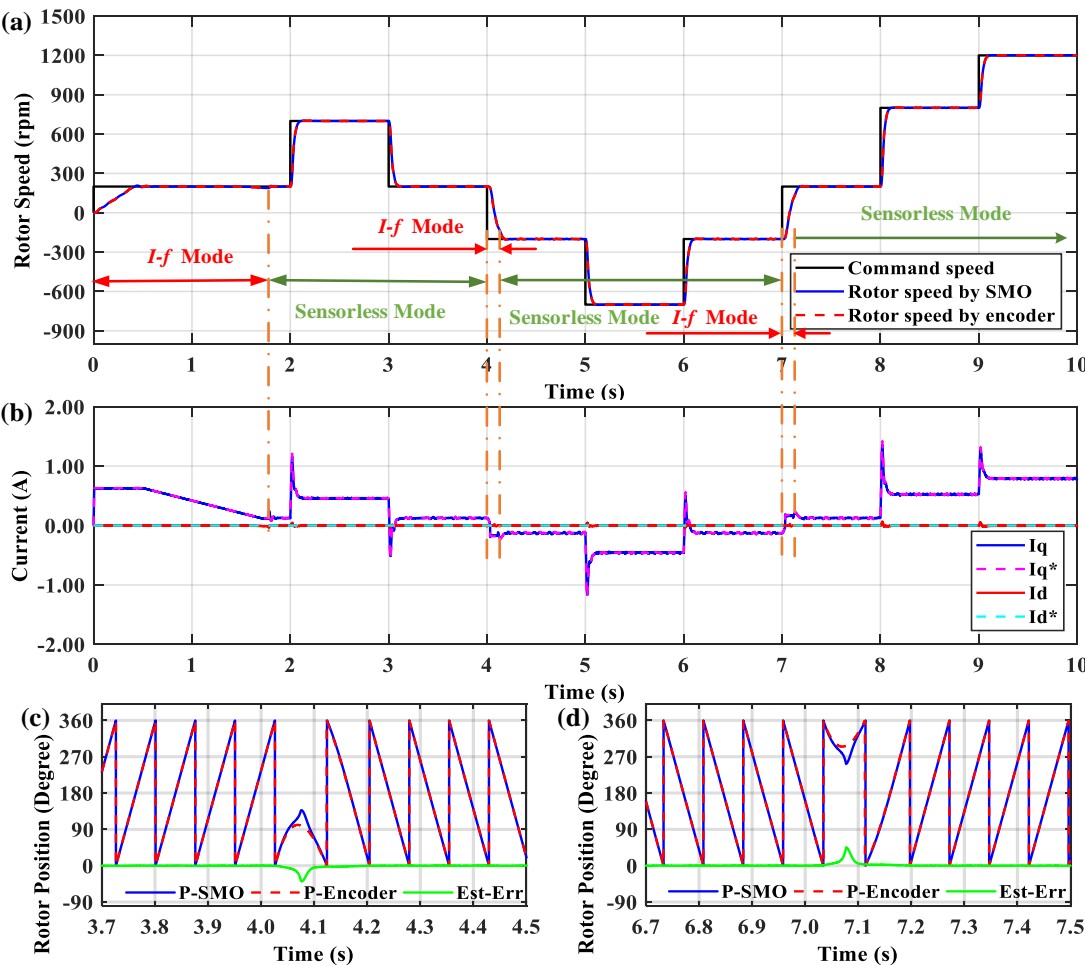

**Figure 10.** Simulation results in the case of the varied rotational direction for: (**a**) speed response; (**b**) current response; (**c**) rotor position (positive–negative); and (**d**) rotor position (negative–positive).

The system performance for the PI speed controller and the NFC-based speed controller is presented in Figure 11 when the motor startups with the initial resistance load of 100 Ω. The rotor speed follows the sequence of 0 → 200 → 500 → 1000 rpm until t = 4 s. Then, the command speed is regulated as a square wave with a period of 1 s and the speed variation from 1000 to 1500 rpm. At t = 5.5 s, more resistance load is added to the total value of 50 Ω. It implies that the larger external load is suddenly applied to the motor, resulting in a drop of rotor speed briefly because it takes some time to raise the energy supplied to the system. In the PI controller, the rotor speed decreases to the minimum of 950 rpm at t = 5.531 s and steadies at 1000 rpm again at t = 5.751 s. The recovery time is 0.220 s with a speed reduction of 50 rpm. In the NFC controller, the rotor speed decreases to the minimum of 964 rpm at t = 5.523 s and steadies at 1000 rpm again at t = 5.679 s. The recovery time is 0.156 s with a speed reduction of 36 rpm.

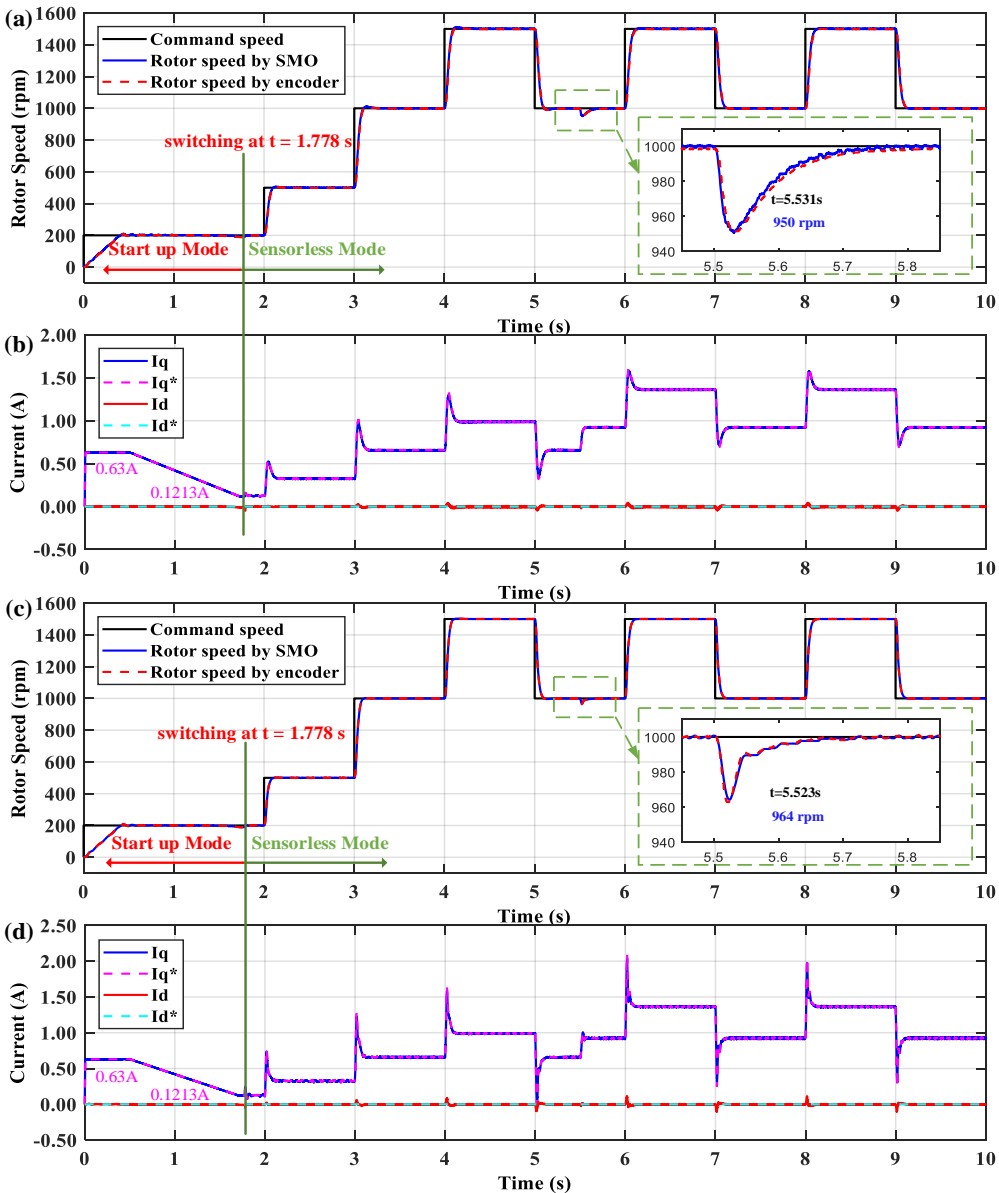

**Figure 11.** Simulation results for PI (Proportional-Integral) controller: (**a**) speed response; and (**b**) current response; and for NFC controller: (**c**) speed response; and (**d**) current response in the case of $R_L = 100 \to 50$ Ω.

Figure 12 shows the system performance when the motor startups with the initial resistance load of 50 Ω. The speed command is also varied as the same as its waveform, as shown in Figure 11. Because the larger external load is applied in the startup mode, before the earlier sensorless control mode switching at t = 1.702 s, the reference current $i_q^*$ is reduced to the value of 0.1818 A (Figure 12), which is higher than the case shown in Figure 11. Those corresponding values in Figure 11 are 1.778 s and 0.1213 A, respectively. At t = 5.5 s, the resistance load is varied to obtain the total value of 100 Ω. It implies that less external load is applied to the system. In the PI controller, the motor increases the speed until reaching the maximum speed of 1051 rpm, at t = 5.526 s, and then stabilizes at 1000 rpm again at t = 5.724 s. The recovery time is 0.198 s with the speed increment of 51 rpm. In the NFC controller, the rotor speed is increased to the maximum of 1025 rpm, at t = 5.519 s, and stabilizes at 1000 rpm again at t = 5.687 s. The recovery time is 0.168 s with the speed increment of 25 rpm.

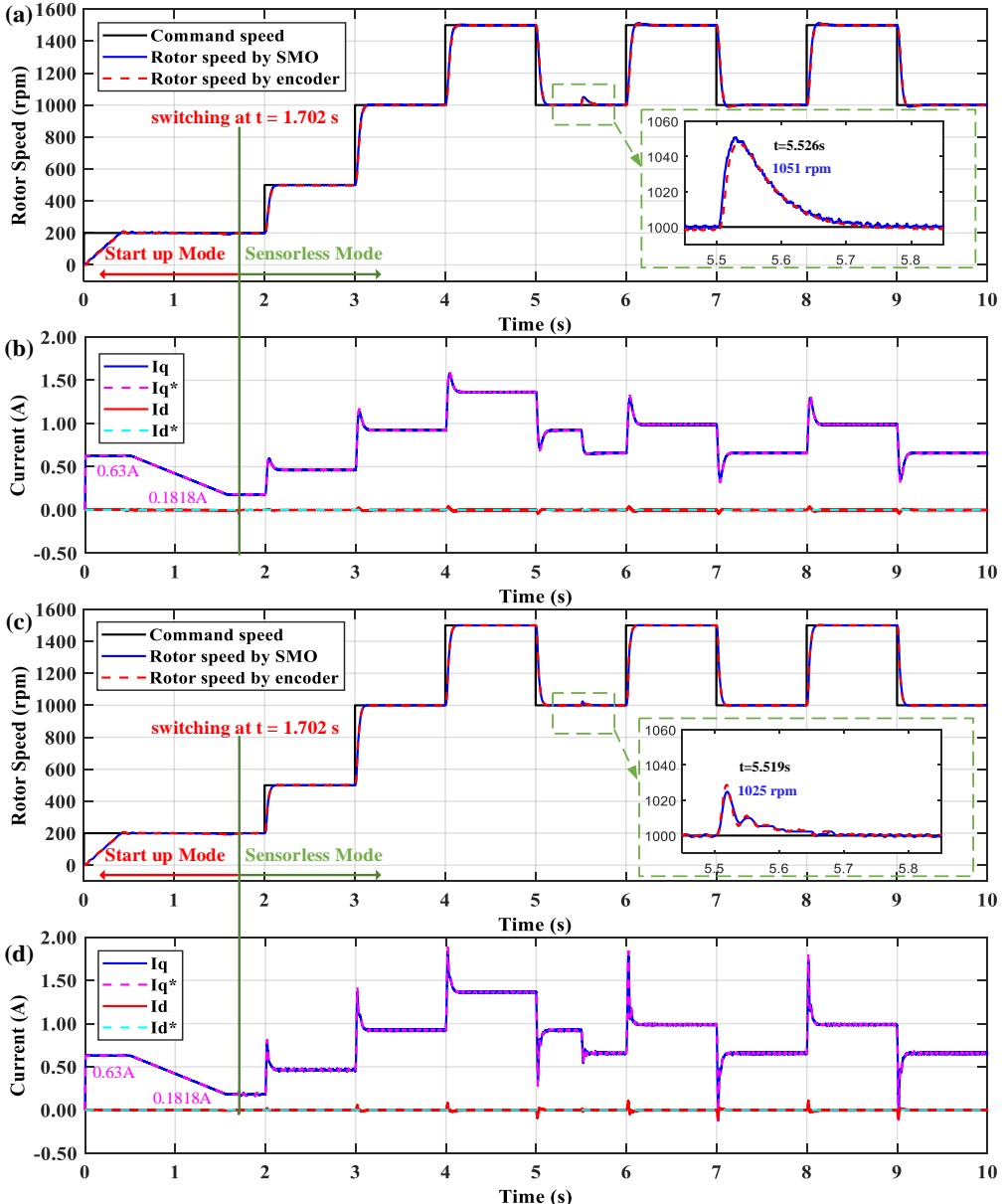

**Figure 12.** Simulation results for PI controller: (**a**) speed response; and (**b**) current response; and for NFC controller: (**c**) speed response; and (**d**) current response in the case of R$_L$ = 50 → 100 Ω.

The comparison of the different speed performances for the PI speed controller and the NFC-based speed controller is presented in Figures 13 and 14. It is easy to see that the system has a better performance in the NFC-based speed controller. The motor speed tracks the command speed perfectly, without the overshoots or undershoots for two cases of external load. It also has a faster response, with the settling time of 0.149 s. However, there is a little difference in the speed performance for two cases of external load in the PI speed controller. In the case of $R_L = 100 \ \Omega$, the system has an overshoot and an undershoot, at about +10 rpm, and the settling time of 0.267 s. The system has better performance, without overshoots or undershoots, and the settling time of 0.164 s in the case of $R_L = 50 \ \Omega$, where the PI's parameters are set appropriately. Figures 13 and 14 demonstrate that the PI speed controller can only work properly at the defined condition ($R_L = 50 \ \Omega$), while the proposed NFC-based speed controller works effectively for both cases to obtain a good performance because it has a mechanism to adjust its parameters, adapted to the dynamic system characteristic.

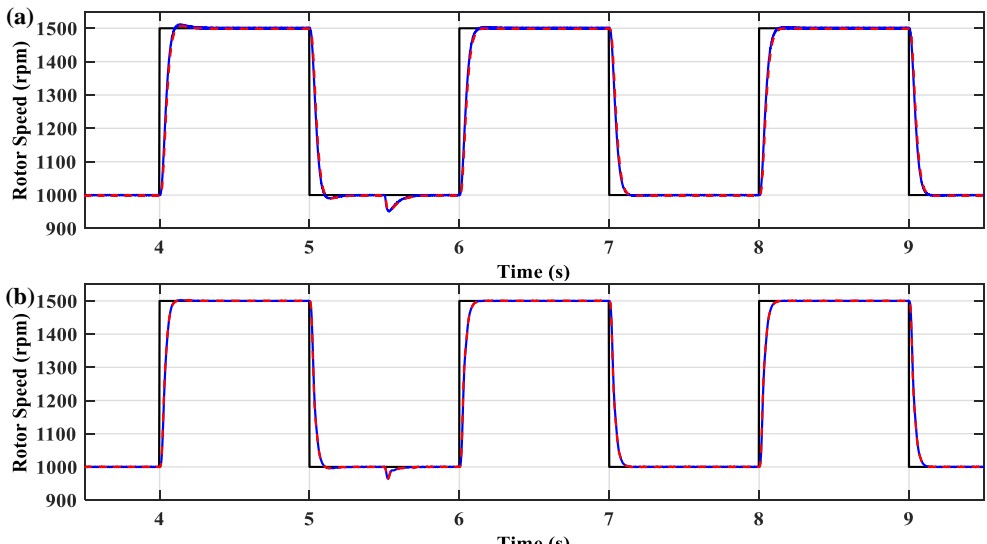

**Figure 13.** Comparison of the speed response of simulation results in the case of the varied external load for $R_L = 100 \to 50 \ \Omega$: (**a**) PI controller; and (**b**) NFC controller.

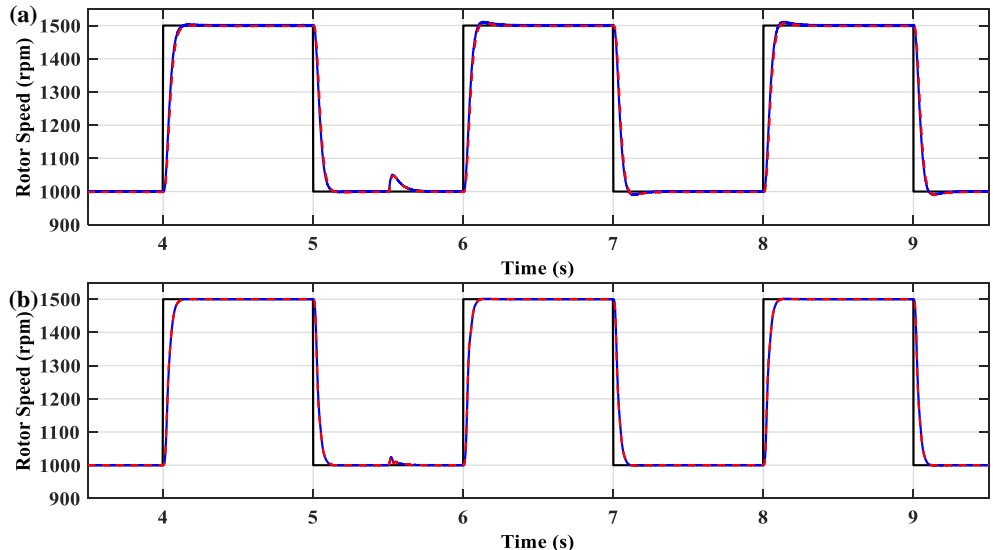

**Figure 14.** Comparison of the speed response of simulation results in the case of the varied external load for $R_L = 50 \to 100 \ \Omega$: (**a**) PI controller; and (**b**) NFC controller.

In summary, the motor control system's performance is analyzed in Figures 8–14. Several command speed waveforms and the varied dynamic load conditions are applied. Those results imply that the estimator works successfully. The estimated values approach the actual values for both rotor speed and rotor position. The NFC-based speed controller improves the system performance effectively and presents the robustness against disturbance. The rotor speed almost tracks the command perfectly, with a zero steady-state error, without overshoot or undershoot, and having shorter settling time, in comparison to the PI speed controller. Additionally, the system switches the control mode smoothly and reverses the rotational direction stably when the *I-f* control strategy is implemented. The simulation results have already confirmed that the proposed control algorithm for the sensorless PMSM drive control system is correct and effective.

## 5. Experimental Verification and Results

After evaluating the system performance by simulation, the motor control algorithm was compiled and deployed to the experimental PMSM drive control system, as presented in Figure 15. Moreover, to analyze the system performance, the system data were transmitted to MATLAB Simulink by the SCI function, integrated into the DSP with the sampling frequency of 2 kHz. The experimental system involves a PMSM coupled to a generator, an inverter, a control circuit, and a DSP F28379D. The DSP F28379D is equipped with 200 MHz dual C28xCPUs and dual CLAs, 1 MB Flash, 16-bit/12-bit ADCs, comparators, 12-bit DACs, HRPWMs, eCAPs, eQEPs, CANs, etc. The isolation and protection functions are integrated into the control circuit, which helps lock the PWM control signal in the case of overcurrent problem. The electrical load is comprised of a rectifier, a capacitor of 470 µF and 450 V, and the resistor load bank.

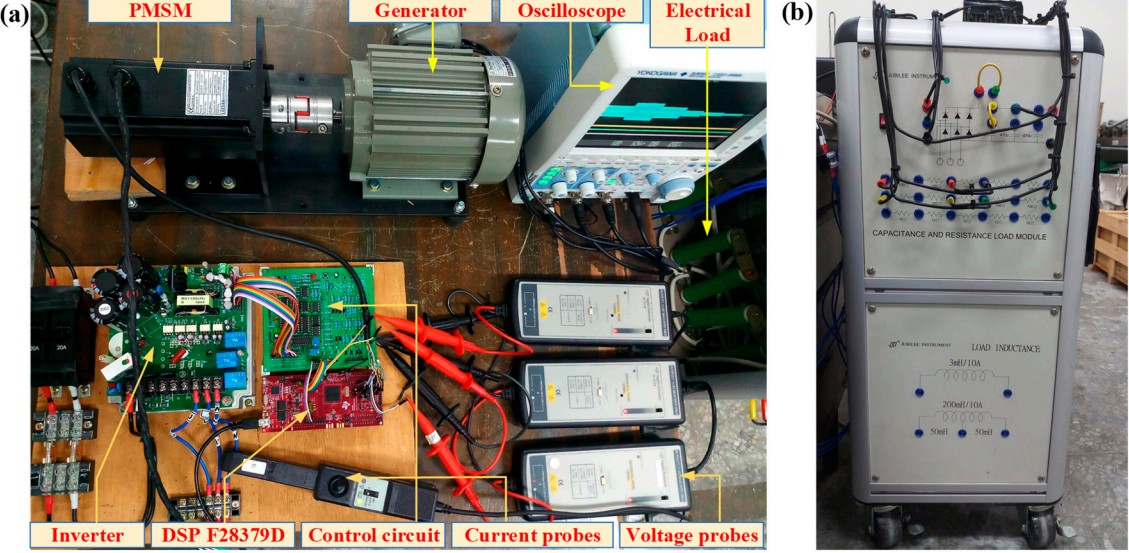

**Figure 15.** The experimental system with: (**a**) real platform; and (**b**) electrical load.

The sampling frequencies of the speed control loop, the current control loop and the SMO-PLL estimator are 1, 20, and 20 kHz, respectively. The inverter's switching frequency is set at 15 kHz. The PI controllers' parameters are set as $K_{Pq}$ = 0.25, $K_{Iq}$ = 0.025, $K_{Pd}$ = 0.25, $K_{Id}$ = 0.025, $K_{pw}$ = 0.735, and $K_{iw}$ = 0.00918. The membership function's values are set the same as the simulation configuration. The fuzzy rule table values are initialized with a ratio of 0.833 to the values in Table 1 and adjusted with the adaptive rate of 0.25. The neural network's parameters are set as follows: node centers $C_l$ = [−2.5 −1.25 0.0 1.25 2.5], node widths $b$ = [2.5 2.5 2.5 2.5 2.5], connective weights $w$ = [0.025 0.025 0.025 0.025 0.025], momentum factor $\alpha$ = 0.75, and learning rate $\eta$ = 0.435. In the first and second experimental conditions, the motor is operated with the total resistance load of 100 Ω and 400 W.

Figure 16 presents the motor's performance in the positive direction for the wide speed range. The motor startups from the standstill stage at t = 1.193 s and switches to the sensorless control mode at t = 9.213 s. The rotor speed is accelerated to the rated speed and then decelerated. The command speed is varied in a sequence of 0 → 300 → 500 → 1000 → 1500 → 2000 → 1600 → 1200 → 800 → 1000 rpm for each period of 5 s. Figure 16a refers that the estimated rotor speed overlaps the actual rotor speed and closely tracks the command speed perfectly. In Figure 16b, although the current $i_q$ fluctuates around the reference current $i_q^*$, its average value is still regulated proportionally to the command speed while the current $i_d$ almost oscillates around the zero value. Figure 8c–f illustrates the estimated position, actual position, and the estimation errors at the speeds of 500, 1000, 1500, and 2000 rpm in a period of 0.15 s. There are 5, 10, 15 and 20 position cycles corresponding to the rotation frequencies of 33.33, 66.67, 100, and 133.33 Hz, respectively. These values are proper to the motor with four pole pairs. The actual and estimated positions come close to each other; thus, the estimation error approximates zero.

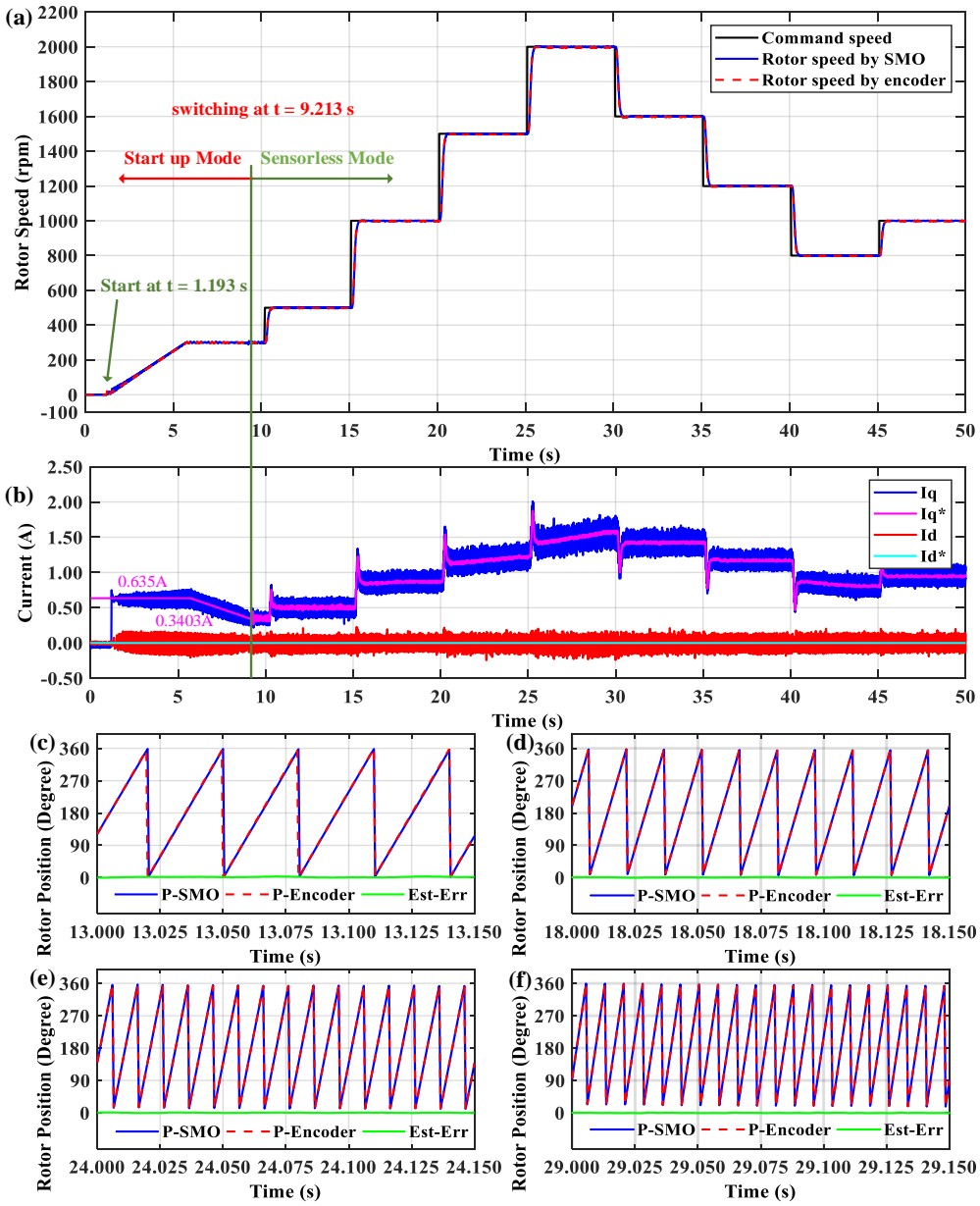

**Figure 16.** Experimental results in the wide speed range in the positive direction: (**a**) speed response; (**b**) current response; and rotor position response at: (**c**) 500; (**d**) 1000; (**e**) 1500; and (**f**) 2000 rpm.

Figure 17 illustrates the motor's performance in the negative direction for the wide speed range. The waveform of the command speed is the same as in the positive direction, only the speed has the opposite value. The motor starts up at t = 1.035 s and switches to the sensorless control mode at t = 9.201 s. Comparing to Figure 16, in the positive direction, the rotor position is increased from 0° to 360° for one electrical cycle and the estimation error is close to zero, while the rotor position is decreased from 360° to 0° in the negative direction and the estimation error is not completely equal to zero; the maximum error values can reach to 12°. Therefore, the reference current $i_q^*$ is regulated with a little larger value, in comparison to the case of the positive direction. However, the rotor position still tracks the command speed very well. In both rotational directions, the motor starts up from the standstill stage with the initial reference current $i_q^*$ of 0.635 A. The ratio for decreasing the reference current $i_q^*$ is set up at the value of 0.085 A/s.

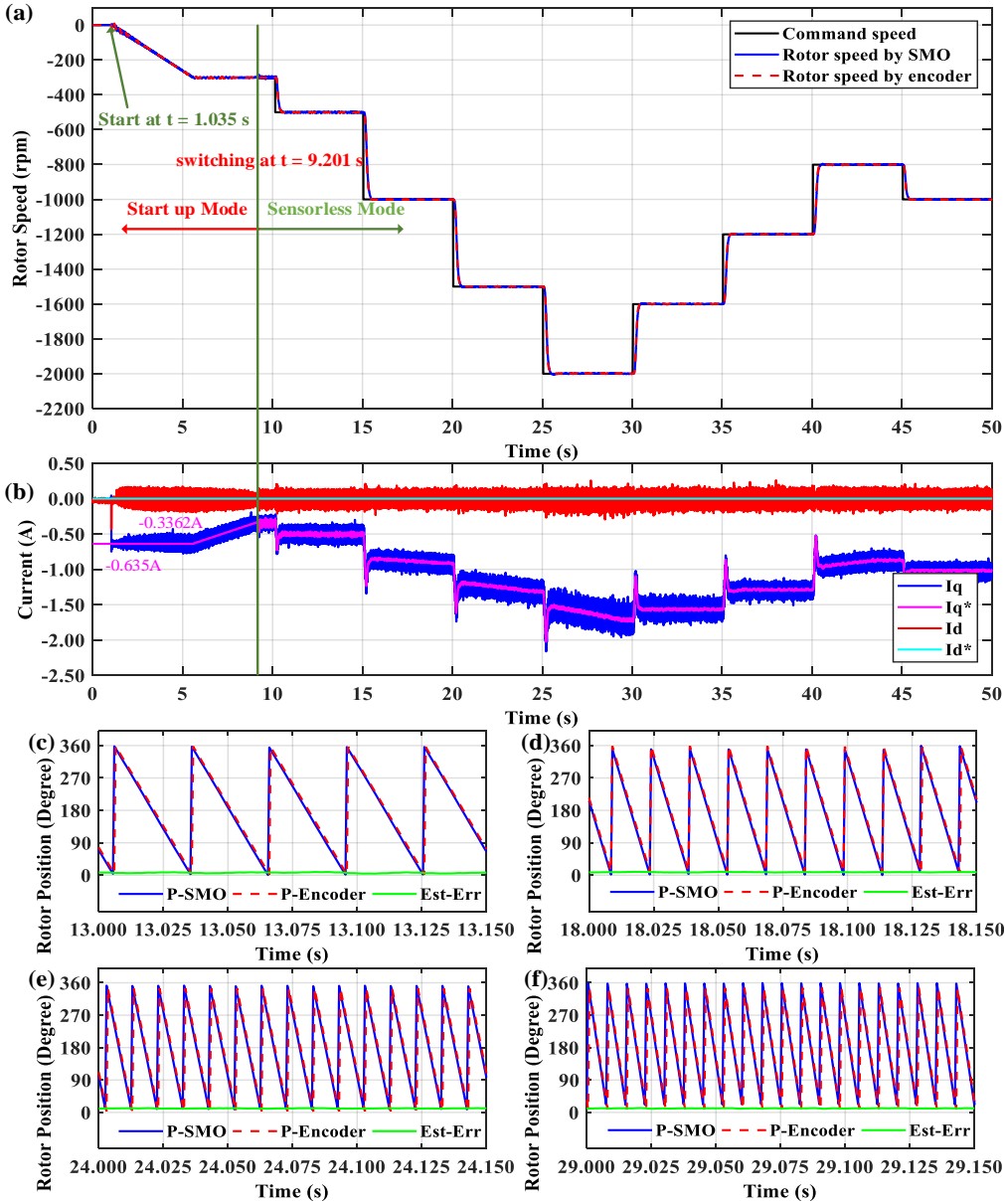

**Figure 17.** Experimental results in the wide speed range in the negative direction: (**a**) speed response; (**b**) current response; and rotor position response at: (**c**) 500; (**d**) 1000; (**e**) 1500; and (**f**) 2000 rpm.

The motor's speed performance in the reversal operation is presented in Figure 18. Firstly, the motor rotates in the negative direction, and the rotor speed is varied in the sequence of −300 →

−700 → −300 rpm. Secondly, the command speed is set up to reverse the rotational direction from −300 to +300 rpm at t = 11 s. The motor control algorithm is transferred to the *I-f* control mode from the sensorless control mode. The motor is decelerated to zero and accelerated to the command speed, following a ramp function with a ratio of 266.67 rpm/s. The actual rotor speed reaches 300 rpm at t = 13.28 s. It takes about 2.28 s to reverse the rotational direction from the negative direction to the positive direction. When the motor has already operated in the positive direction, the motor control algorithm is switched to the sensorless control mode again at t = 14.79 s. The *I-f* control mode is implemented in a period of 3.79 s in this direction reversion transition. Then, the command speed is varied by the sequence: 300 → 700 → 1200 → 300 rpm. Thirdly, a similar procedure is applied when the command speed is changed to reverse the rotational direction from +300 to −300 rpm at t = 36 s. The actual rotor speed reaches −300 rpm at t = 38.26 s. It takes about 2.26 s to reverse the rotational direction from the positive direction to the negative direction. The *I-f* control mode is active in a period of 4.02 s. During the operation of the motor, the estimated rotor speed almost tracks the command speed and approximates the actual rotor speed for both the sensorless control mode and the *I-f* control mode. Moreover, the current response in Figure 18b indicates that the reference current $i_q^*$ is regulated differently for the sensorless control mode and the *I-f* control mode in the case of the rotational direction reversion. Therefore, the rising time and settling time are different in the speed transition between the rotational direction reversion and the uni-direction speed variation. It takes a larger time to reverse the rotational direction. In the first direction reversion transition, the current $i_q^*$ is regulated from −0.3652 to 0.6226 A, and then reduced to 0.3362 A before switching to the sensorless control mode. In the second transition, those corresponded values are 0.3694, −0.6558, and −0.3237 A, respectively.

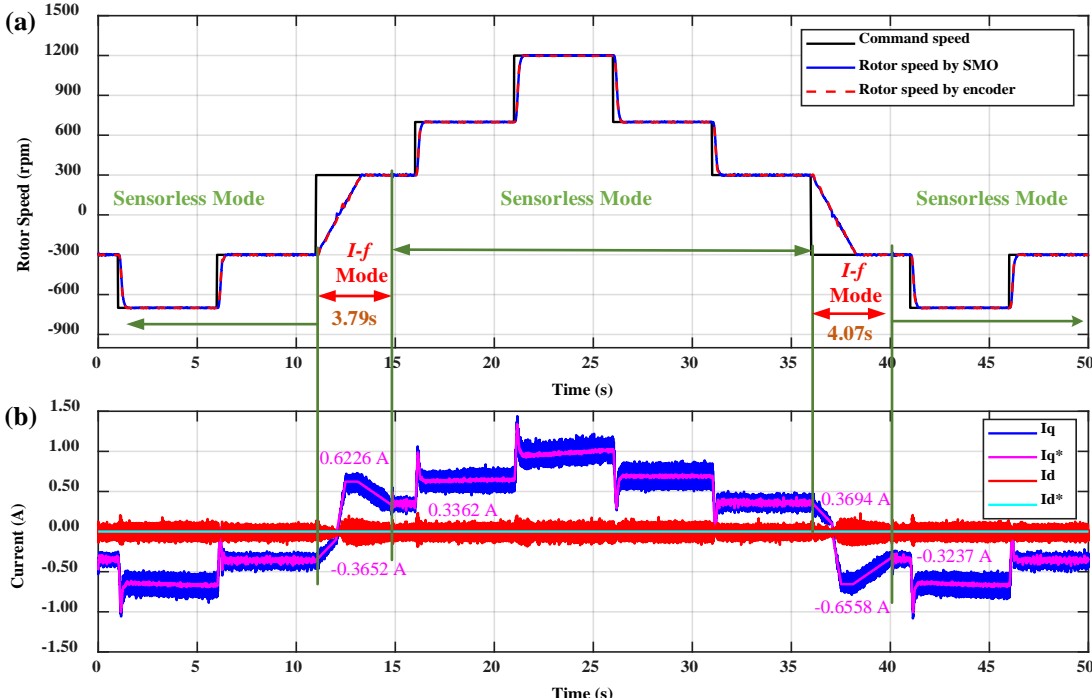

**Figure 18.** Experimental results in the case of the varied rotational direction for: (**a**) speed response; and (**b**) current response.

Figure 19a–c illustrates the rotor position when the motor changes the rotational direction to the positive direction. When the actual speed crosses the zero value, there is a large estimation error because the modified PLL changes the parameter $\theta_{offset}$ from 180° to 0°. However, this error is reduced significantly when the system is switched to the sensorless control mode. In Figure 19e–f, a similar result is analyzed for rotor position in the case of changing the rotor speed from the positive value to negative value. The parameter $\theta_{offset}$ is transferred from 0° to 180°. Finally, Figures 18 and 19 verify

that the *I-f* control mode makes the motor change the rotational direction stably and the modified conventional PLL is worked effectively, which is not affected by noise, inaccurate back-EMF estimation or large estimation error in the low-speed range. The estimated position still approaches the actual value after the direction reversion transition has occurred.

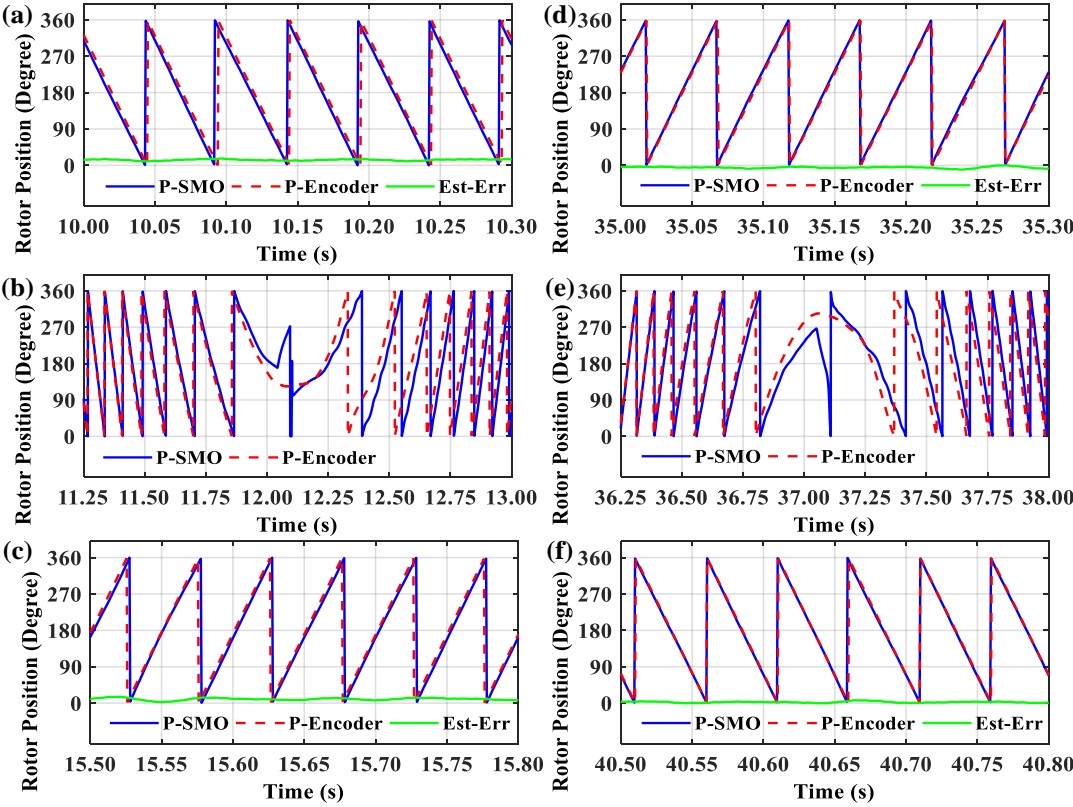

**Figure 19.** Experimental results of rotor position in the case of the varied rotational direction: from negative to positive (**a**–**c**); and from positive to negative (**d**–**f**).

Figure 20 presents the system performance for the PI speed controller and the NFC-based speed control when the motor startups with the initial resistance load of 100 Ω and 400 W. The motor startups at t = 2.821 s and switches to the sensorless control mode at t = 10.915 s. The rotor speed follows the sequence of 0 → 300 → 500 → 1000 rpm. Then, the command speed is regulated as a square wave with a period of 5s and the speed variation from 1000 to 1500 rpm. In the PI controller, at t = 28.94 s, more power resistors are added to increase the external load, and the total resistance load transfers to the new value of 50 Ω and 800 W. At t = 29.06 s, the rotor speed reduces to the low peak, with the estimated value (blue line) of 878 rpm and the actual value (red line) of 867 rpm. The rotor speed steadies at 1000 rpm again at t = 29.84 s. The recovery time is 0.78 s with the actual speed reduction of 133 rpm. In the NFC controller, the external load is enhanced at t = 30.00 s. The rotor speed reduces to the low peak, with the estimated value of 930 rpm and the actual value of 914 rpm at t = 30.09 s. The rotor speed steadies at 1000 rpm again at t = 30.67 s. The recovery time is 0.58 s with the actual speed reduction of 86 rpm.

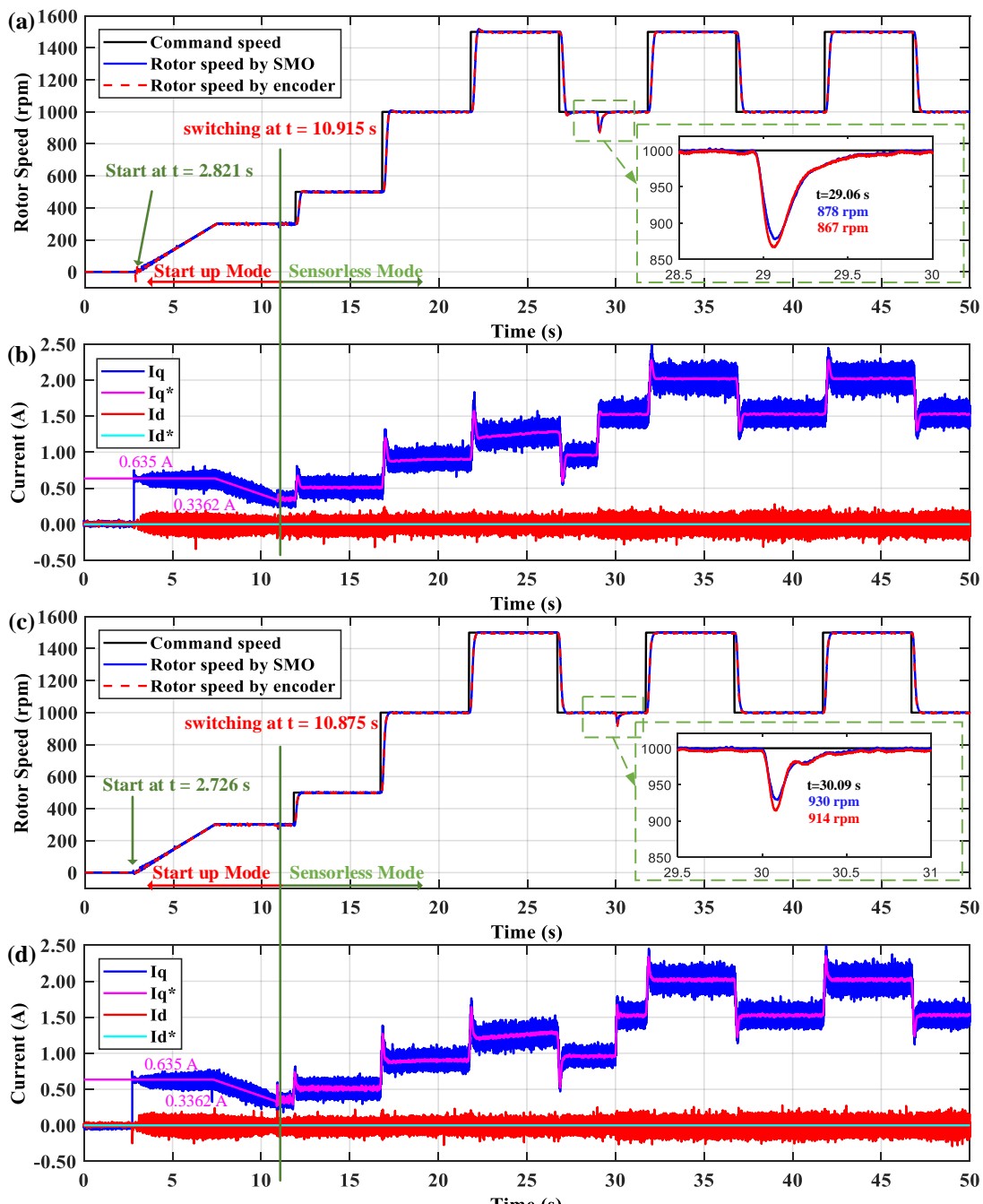

**Figure 20.** Experimental results for PI controller: (**a**) speed response; (**b**) current response; and for NFC controller: (**c**) speed response; and (**d**) current response in the case of $R_L = 100 \rightarrow 50 \, \Omega$.

By the initial resistance load of 50 $\Omega$ and 800 W, the system performance for the PI speed controller and the NFC-based speed controller is shown in Figure 21. The speed command is the same as its waveform presented in Figure 20. Comparing to the results in Figure 20, because a larger external load is applied in the startup mode, the reference current $i_q^*$ is ramped down to 0.5063 A, and then the system switches to the sensorless control mode in a shorter time. It takes a period of 6.087 s (for NFC) in the *I-f* control mode. The corresponding values in Figure 20 are 0.3362 A and 8.094 s, respectively. In the PI controller, at t = 28.08 s, some power resistors are removed to reduce the external load, and the total resistance load varies to the new value of 100 $\Omega$ and 400 W. At t = 28.19 s, the rotor speed increases to the up peak, with the estimated value of 1132 rpm and the actual value of 1141 rpm. The rotor speed steadies at 1000 rpm again at t = 28.89 s. The recovery time is 0.70 s with the actual speed increment of

141 rpm. In the NFC controller, the external load is reduced at t = 28.92 s. The rotor speed goes to the up peak with the estimated value of 1072 rpm and the actual value of 1083 rpm at t = 29.00 s. The rotor speed stabilizes at 1000 rpm again at t = 29.50 s. The recovery time is 0.50 s with the actual speed increment of 83 rpm. Lastly, Figures 20 and 21 indicate that the sensorless motor control algorithm still works stably and successfully, robust to the disturbance of the external load.

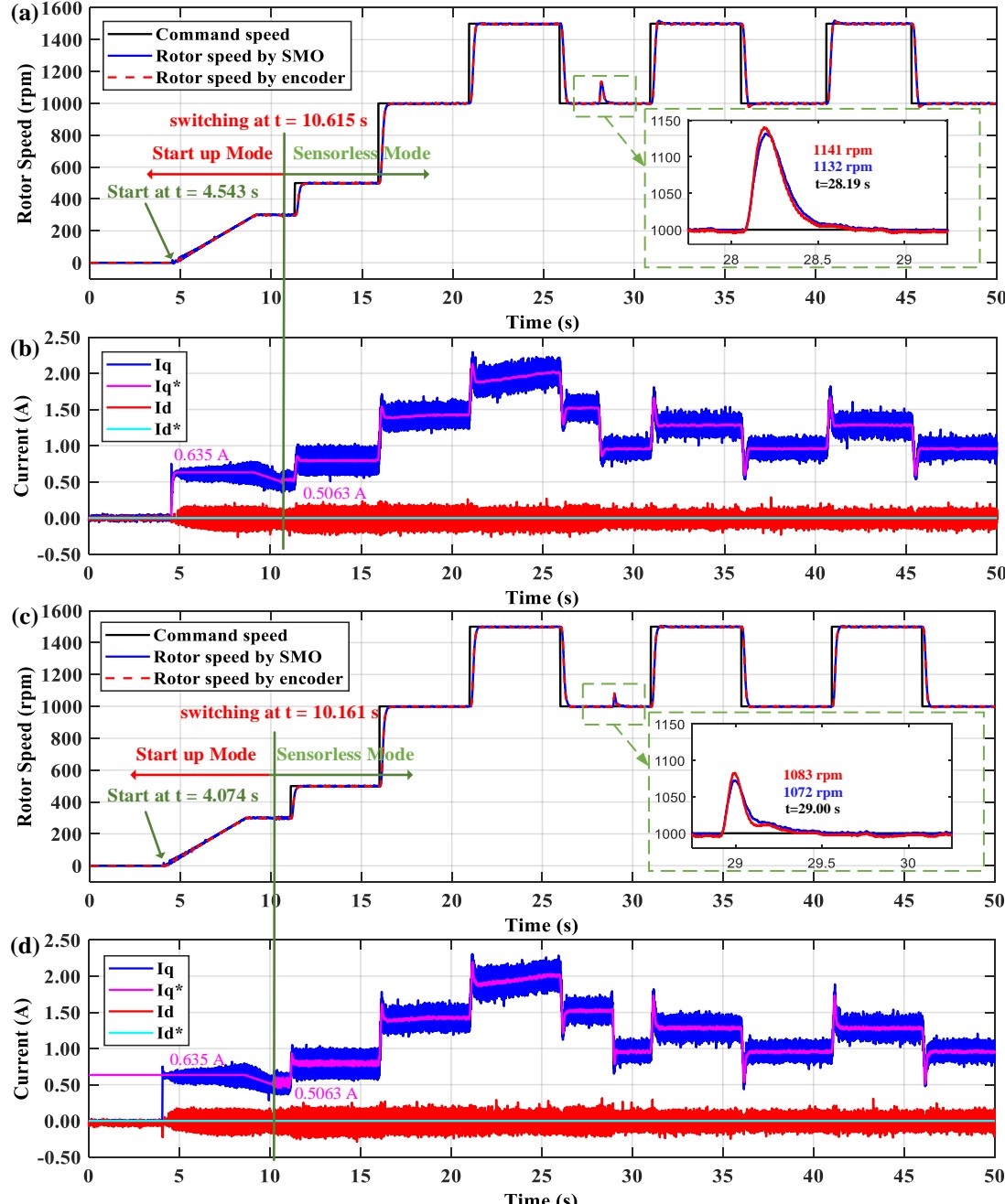

**Figure 21.** Experimental results for PI controller: (**a**) speed response; (**b**) current response; and for NFC controller: (**c**) speed response; and (**d**) current response in the case of the $R_L = 50 \rightarrow 100 \ \Omega$.

The comparison of the detailed speed performances for the PI speed controller and the NFC-based speed controller is presented in Figures 22 and 23. This demonstrates that the NFC-based speed controller creates better performance. The motor speed tracks the command speed perfectly, without the overshoots or undershoots for two cases of external load and the settling time of 0.50 s. However,

there is a little difference in the speed performance for the PI speed controller. In the case of $R_L = 100\ \Omega$ and 400 W, the system has an overshoot and an undershoot, at about +17 rpm, and the settling time of 0.78 s. While the system has a good performance, without overshoots or undershoots, and the settling time of 0.53 s in the case of $R_L = 50\ \Omega$ and 800 W. Figures 22 and 23 prove again that the PI speed controller can only work properly at the specific condition ($R_L = 50\ \Omega$ and 800 W), while the proposed NFC-based speed controller works effectively for both cases to obtain a good performance because it has a mechanism to adjust its parameters, adapting to the dynamic system characteristic.

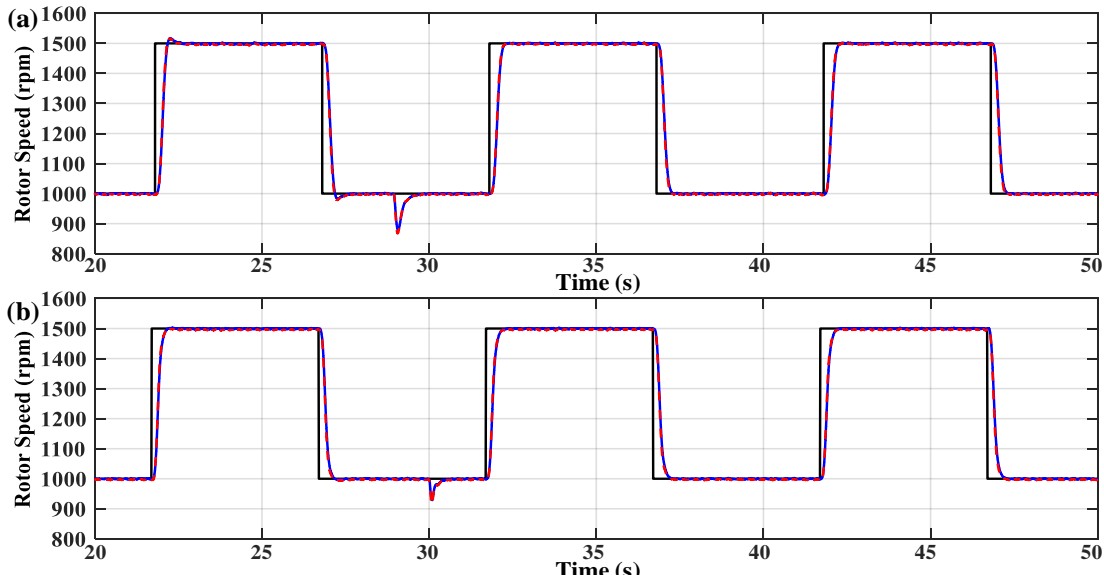

**Figure 22.** Comparison of speed response of experimental results in the case of the varied external load for $R_L = 100 \to 50\ \Omega$: (**a**) PI controller; and (**b**) NFC controller.

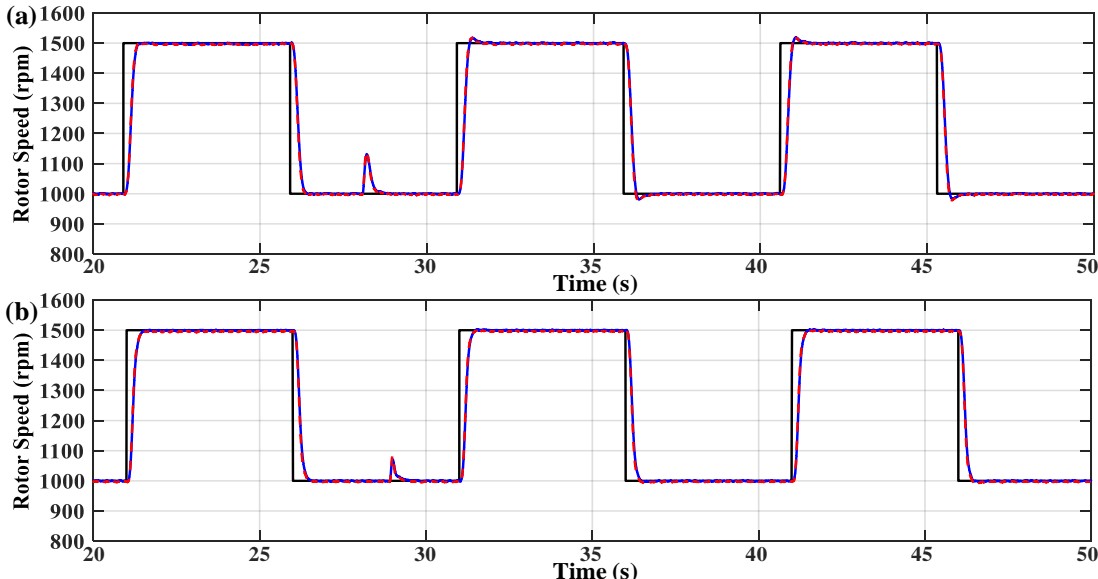

**Figure 23.** Comparison of speed response of experimental results in the case of the varied external load for $R_L = 50 \to 100\ \Omega$: (**a**) PI controller; and (**b**) NFC controller.

In summary, the experimental results in Figures 16–23 verify that the proposed control algorithm for the sensorless PMSM drive control system is realized effectively. The motor can startup with different initial external load and switch to the sensorless control mode smoothly. Furthermore,

the motor operates in two rotational directions and the direction reversion transition is executed stably. It confirms that the estimator is implemented successfully in combination with the *I-f* control strategy. The estimated values approach the actual value for both rotor speed and rotor position. Moreover, compared to the PI speed controller, the NFC-based speed controller improves the system performance excellently and presents robustness against disturbance. The rotor speed tracks the command properly, without the overshoot or undershoot. The steady-state error almost comes close to zero (within ±5 rpm in the tolerance). Additionally, the DSP application for the PMSM drive control system is properly designed in MATLAB Simulink, deployed to Code Composer Studio software (Version 8.0.0.00016, Texas Instruments, Inc., Dallas, TX, USA) and successfully realized in the real-time system. Therefore, the development time for DSP application is substantially shortened by this deployment method.

## 6. Conclusions

In this work, the NFC-based speed controller and SMO-PLL estimator for the sensorless PMSM drive control system are presented and realized effectively. The proposed control algorithm was designed in MATLAB Simulink and deployed to the real-time platform, based on a DSP F28379D. Different experimental conditions were executed to evaluate system performance. The combination of the SMO-PLL estimator and the *I-f* control strategy eliminates the initial rotor position estimation and overcomes the reversal problem. The motor can startup with a diverse external load, operate in two directions in a wide speed range, and switch the rotational direction stably. The analyzed results demonstrate that the estimator works correctly. The estimated values come close to the actual value for both rotor position and speed so that the estimation error is almost insignificant. Furthermore, the NFC-based speed controller improves the system performance outstandingly and presents robustness against disturbance. The rotor speed almost tracks the command perfectly, with a zero steady-state error, without overshoot or undershoot, and having shorter settling time in comparison to the PI speed controller. Therefore, the system performance demonstrates that the proposed control algorithm for the sensorless PMSM drive control system is correct and effective.

**Author Contributions:** H.-K.H. wrote this article, designed the control method, implemented the hardware platform, drew the figures, and performed the simulations as well as the experiments. S.-C.C. supervised and coordinated the investigations, reviewed the manuscript, and checked its logical structure. C.-F.C. supplied the materials and reviewed the article draft. All authors have read and agreed to the published version of the manuscript.

**Funding:** This research was funded by Fukuta Electric & Machinery Co., Ltd, Taiwan, under contract No. 14001080308.

**Conflicts of Interest:** The authors declare no conflict of interest.

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
