# Peer review of "Realization of the Neural Fuzzy Controller for the Sensorless PMSM Drive Control System"

_electronics, doi:10.3390/electronics9091371_

Round 1
Reviewer 1 Report
A neural fuzzy controller (NFC)-based speed controller for the sensorless PMSM drive control system is proposed in this paper. The proposed strategy has been clearly presented.
However, I still have one concern about the simulation and experimental results. For a fair comparison with the proposed strategy, it is important to discuss how to choose the parameters of the PI-based speed controller. The authors are suggested to be provided the fair selection rules in the modified version. Besides, in line 15, the abbreviation of SPMSM is not given.
Reviewer 2 Report
The manuscript “Realization of the Neural Fuzzy Controller for the Sensorless PMSM Drive Control System” discusses a neural fuzzy controller (NFC)-based speed controller for the sensorless permanent magnet synchronous motor (PMSM) drive control system. The NFC is a fuzzy logic controller (FLC), which is adaptively to adjust the RBFNN-based (radial basis function neural network) parameter adapting the dynamic system characteristics. The manuscript is well written and logically set out with satisfactory quality of figures and results for both simulations and experiments. It is this reviewers opinion that the work be is suitable for publication after some minor corrections. Suggestions are below:
- The reviewer recommends another English proofreading as there are still some minor grammatical errors.
- While the topic of NFC in motor control with sliding mode observers is very interesting, there have already been many works in this area. To this end, the authors may make comparisons to previous work or clearly explain in the introduction what is particularly new about their own work.
- In Figures 19 b and 3, there is large discrepancy in the SMO and encoder with what appears to be sharp discontinuities in the SMO. How have the authors addressed this issue as this is severely detrimental in control applications.
- The reviewer suggests that the authors modify the manuscript to shorten its length in order to be more precise. Whilst many error equations were given, the authors should only use equations that are relevant to their current work and substantially reduce the Figure count.
